# Conformal confidence sets for biomedical image segmentation

## Abstract

We develop confidence sets which provide spatial uncertainty guarantees for the output of a black-box machine learning model designed for image segmentation. To do so we adapt conformal inference to the imaging setting, obtaining thresholds on a calibration dataset based on the distribution of the maximum of the transformed logit scores within and outside of the ground truth masks. We prove that these confidence sets, when applied to new predictions of the model, are guaranteed to contain the true unknown segmented mask with desired probability. We show that learning appropriate score transformations on an *independent* learning dataset before performing calibration is crucial for optimizing performance. *We illustrate and validate our approach on polyps colonscopy, brain imaging and teeth datasets. To do so we obtain the logit scores from deep neural networks trained for polyps, brain mask and tooth segmentation segmentation. We show that using distance and other transformations of the logit scores allows us to provide tight inner and outer confidence sets for the true masks whilst controlling the false coverage rate.*

## 1 Introduction

Deep neural networks promise to significantly enhance a wide range of important tasks in biomedical imaging. However these models, as typically used, lack formal uncertainty guarantees on their output which can lead to overconfident predictions and critical errors (Guo et al., 2017; Gupta et al., 2020). Misclassifications or inaccurate segmentations can lead to serious consequences, including misdiagnosis, inappropriate treatment decisions, or missed opportunities for early intervention (Topol, 2019). Without uncertainty quantification, medical professionals cannot rely on deep learning models to provide accurate information and predictions which can limit their use in practical applications (Jungo et al., 2020).

In order to address this problem, conformal inference, a robust framework for uncertainty quantification, has become increasingly used as a means of providing prediction guarantees, offering reliable, distribution-free confidence sets for the output of neural networks which have finite sample validity. This approach, originally introduced in Papadopoulos et al. (2002); Vovk et al. (2005), has become increasingly popular due to its ability to provide rigorous statistical guarantees without making strong assumptions about the underlying data distribution or model architecture. Conformal prediction methods, in their most commonly used form - split conformal inference - work by calibrating the predictions of the model on a held-out dataset in order to provide sets which contain the output with a given probability, see Shafer & Vovk (2008) and Angelopoulos & Bates (2021) for good introductions.

In the context of image segmentation, we have a decision to make at each pixel/voxel of an image which can lead to a large multiple testing problem. Traditional conformal methods, typically designed for scalar outputs, require adaptation to handle multiple tests and their inherent spatial dependencies. To do so Angelopoulos et al. (2021) applied conformal inference pixelwise and performed multiple testing correction on the resulting $p$-values, however this approach does not account for the complex dependence structure inherent in the images. To take advantage of this structure, in an approach analogous to the *False discovery rate (FDR)* control of (Benjamini & Hochberg, 1995), Bates et al. (2021) and Angelopoulos et al. (2024) sought to control the expected risk of a given loss function over the image and used a conformal approach to produce outer confidence sets for

segmented images which control the expected proportion of false negatives. Other work considering conformal inference in the context of multiple dependent hypotheses includes Marandon (2024) and Blanchard et al. (2024) who established conformal FDR control when testing for the presence of missing links in graphs.

In this work we argue that bounding the segmented outcome with guarantees in probability rather than on the proportion of discoveries is more informative, avoiding errors at the borders of potential growths/tumors. This is analogous to the tradeoff between *familywise error rate (FWER) and FDR* control in the multiple testing literature in which there is a balance between power and coverage rate, *(a correspondence which we formalize in Section A.10)*. The distinction is that in medical image segmentation making mistakes can have potentially serious consequences Under-segmentation might cause part of the true mask to be missed, potentially leading to inadequate treatment (Jalalifar et al., 2022). Over-segmentation, on the other hand, could result in unnecessary interventions, increasing patient risk and healthcare costs (Gupta et al., 2020; Patz et al., 2014). Confidence sets are instead guaranteed to contain the outcome with a given level of certainty. Since the guarantees are more meaningful the problem is more difficult and existing work on conformal uncertainty quantification for images has thus often focused on producing sets with guarantees on the proportions of discoveries or pixel level inference rather than coverage (Bates et al. (2021), Wieslander et al. (2020), Mossina et al. (2024)) which is a stricter error criterion.

In order to obtain confidence sets we use a split-conformal inference approach in which we learn appropriate cutoffs, with which to threshold the output of an image segmenter, from a calibration dataset. These thresholds are obtained by considering the distribution of the maximum logit (transformed) scores provided by the model within and outside of the ground truth masks. This approach allows us to capture the spatial nature of the uncertainty in segmentation tasks, going beyond simple pixel-wise confidence measures. By applying these learned thresholds to new predictions, we can generate inner and outer confidence sets that are guaranteed to contain the true, unknown segmented mask with a desired probability. As we shall see, naively using the original logit scores to do so can lead to rather large and uninformative outer confidence sets but these can be greatly improved using distance transformations.

## 2 THEORY

### 2.1 SET UP

Let $\mathcal{V} \subset \mathbb{R}^m$, for some dimension $m \in \mathbb{N}$, be a finite set corresponding to the domain which represents the pixels/voxels/points at which we observe imaging data. Let $\mathcal{X} = \{g : \mathcal{V} \to \mathbb{R}\}$ be the set of real functions on $\mathcal{V}$ and let $\mathcal{Y} = \{g : \mathcal{V} \to \{0, 1\}\}$ be the set of all functions on $\mathcal{V}$ taking the values 0 or 1. We shall refer to elements of $\mathcal{X}$ and $\mathcal{Y}$ as images. Suppose that we observe a calibration dataset $(X_i, Y_i)_{i=1}^n$ of random images, where $X_i : \mathcal{V} \to \mathbb{R}$ represents the $i$th observed calibration image and $Y_i : \mathcal{V} \to \{0, 1\}$ outputs labels at each $v \in \mathcal{V}$ giving 1s at the true location of the objects in the image $X_i$ that we wish to identify and 0s elsewhere. Let $\mathcal{P}(\mathcal{V})$ be the set of all subsets of $\mathcal{V}$. Given a function $f : \mathcal{X} \to \mathcal{X}$, we shall write $f(X, v)$ to denote $f(X)(v)$ for all $v \in \mathcal{V}$.

Let $s : \mathcal{X} \to \mathcal{X}$ be a score function - trained on an independent dataset - such that given an image pair $(X, Y) \in \mathcal{X} \times \mathcal{Y}$, $s(X)$ is a score image in which $s(X, v)$ is intended to be higher at the $v \in \mathcal{V}$ for which $Y(v) = 1$. The score function can for instance be the logit scores obtained from applying a deep neural network image segmentation method to the image $X$. Given $X \in \mathcal{X}$, let $\hat{M}(X) \in \mathcal{Y}$ be the predicted mask given by the model which is assumed to be obtained using the scores $s(X)$.

In what follows we will use the calibration dataset to construct confidence functions $I, O : \mathcal{X} \to \mathcal{P}(\mathcal{V})$ such that for a new image pair $(X, Y)$, given error rates $\alpha_1, \alpha_2 \in (0, 1)$ we have

$$\mathbb{P}\left(I(X) \subseteq \{v \in \mathcal{V} : Y(v) = 1\}\right) \geq 1 - \alpha_1, \tag{1}$$

$$\text{and } \mathbb{P}\left(\{v \in \mathcal{V} : Y(v) = 1\} \subseteq O(X)\right) \geq 1 - \alpha_2. \tag{2}$$

Here $I(X)$ and $O(X)$ serve as inner and outer confidence sets for the location of the true segmented mask. Their interpretation is that, up to the guarantees provided by the probabilistic statements (1) and (2), we can be sure that for each $v \in I(X)$, $Y(v) = 1$ or that for each $v \notin O(X)$, $Y(v) = 0$. Joint control over the events can also be guaranteed, either via sensible choices of $\alpha_1$ and $\alpha_2$ or by using the joint distribution of the maxima of the logit scores - see Section 2.3.

In order to establish conformal confidence results we shall require the following exchangeablity assumption.

**Assumption 1.** Given a new random image pair, $(X_{n+1}, Y_{n+1})$, suppose that $(X_i, Y_i)_{i=1}^{n+1}$ is an exchangeable sequence of random image pairs in the sense that

$$\{(X_1, Y_1), \dots, (X_{n+1}, Y_{n+1})\} =_d \{(X_{\sigma(1)}, Y_{\sigma(1)}), \dots, (X_{\sigma(n+1)}, Y_{\sigma(n+1)})\}$$

for all permutations $\sigma \in S_{n+1}$. Here $=_d$ denotes equality in distribution and $S_{n+1}$ is the group of permutations of the integers $\{1, \dots, n+1\}$.

Exchangeability or a variant is a standard assumption in the conformal inference literature (Angelopoulos & Bates, 2021) and facilitates coverage guarantees. It holds for instance if we assume that the collection $(X_i, Y_i)_{i=1}^{n+1}$ is an i.i.d. sequence of image pairs but is more general and in principle allows for other dependence structures.

## 2.2 Marginal confidence sets

In order to construct conformal confidence sets let $f_I, f_O : \mathcal{X} \to \mathcal{X}$ be inner and outer transformation functions and for each $1 \le i \le n+1$, let $\tau_i = \max_{v \in \mathcal{V}:Y_i(v)=0} f_I(s(X_i), v)$ and $\gamma_i = \max_{v \in \mathcal{V}:Y_i(v)=1} -f_O(s(X_i), v)$ be the maxima of the function transformed scores over the areas at which the true labels equal 0 and 1 respectively. We will require the following assumption on the scores and the transformation functions.

**Assumption 2.** (Independence of scores) $(X_i, Y_i)_{i=1}^{n+1}$ is independent of the functions $s, f_O, f_I$.

Given this we construct confidence sets as follows.

**Theorem 2.1.** (Marginal inner set) Under Assumptions 1 and 2, given $\alpha_1 \in (0, 1)$, let

$$\lambda_I(\alpha_1) = \inf \left\{ \lambda : \frac{1}{n} \sum_{i=1}^{n} 1[\tau_i \le \lambda] \ge \frac{\lceil (1-\alpha_1)(n+1) \rceil}{n} \right\}, \tag{3}$$

and define $I(X) = \{v \in \mathcal{V} : f_I(s(X), v) > \lambda_I(\alpha_1)\}$. Then,

$$\mathbb{P}(I(X_{n+1}) \subseteq \{v \in \mathcal{V} : Y_{n+1}(v) = 1\}) \ge 1 - \alpha_1. \tag{4}$$

*Proof.* Under Assumptions 1 and 2, exchangeability of the image pairs implies exchangeability of the sequence $(\tau_i)_{i=1}^{n+1}$. In particular, $\lambda_I(\alpha_1)$ is the upper $\alpha_1$ quantile of the distribution of $(\tau_i)_{i=1}^{n} \cup \{\infty\}$ and so, by Lemma 1 of Tibshirani et al. (2019), it follows that

$$\mathbb{P}(\tau_{n+1} \le \lambda_I(\alpha_1)) \ge 1 - \alpha_1.$$

Now consider the event that $\tau_{n+1} \le \lambda_I(\alpha_1)$. On this event, $f_I(s(X_{n+1}), v) \le \lambda_I(\alpha_1)$ for all $v \in \mathcal{V}$ such that $Y_{n+1}(v) = 0$. As such, given $u \in \mathcal{V}$ such that $f_I(s(X_{n+1}), u) > \lambda_I(\alpha_1)$, we must have $Y_{n+1}(u) = 1$ and so $I(X_{n+1}) \subseteq \{v \in \mathcal{V} : Y_{n+1}(v) = 1\}$. It thus follows that

$$\mathbb{P}(I(X_{n+1}) \subseteq \{v \in \mathcal{V} : Y_{n+1}(v) = 1\}) \ge \mathbb{P}(\tau_{n+1} \le \lambda_I(\alpha_1)) \ge 1 - \alpha_1.$$

$\square$

For the outer set we have the following analogous result.

**Theorem 2.2.** (Marginal outer set) Under Assumptions 1 and 2, given $\alpha_2 \in (0, 1)$, let

$$\lambda_O(\alpha_2) = \inf \left\{ \lambda : \frac{1}{n} \sum_{i=1}^{n} 1[\gamma_i \le \lambda] \ge \frac{\lceil (1-\alpha_2)(n+1) \rceil}{n} \right\}, \tag{5}$$

and define $O(X) = \{v \in \mathcal{V} : -f_O(s(X), v) \le \lambda_O(\alpha_2)\}$. Then,

$$\mathbb{P}(\{v \in \mathcal{V} : Y_{n+1}(v) = 1\} \subseteq O(X_{n+1})) \ge 1 - \alpha_2. \tag{6}$$

*Proof.* Arguing as in the proof of Theorem 2.1, it follows that $\mathbb{P}(\gamma_{n+1} \le \lambda_O(\alpha_2)) \ge 1 - \alpha_2$. Now on the event that $\gamma_{n+1} \le \lambda_O(\alpha_2)$ we have $-f_O(s(X_{n+1}), v) \le \lambda_O(\alpha_2)$ for all $v \in \mathcal{V}$ such that $Y_{n+1}(v) = 1$. As such, given $u \in \mathcal{V}$ such that $-f_O(s(X_{n+1}), u) > \lambda_I(\alpha)$, we must have $Y_{n+1}(u) = 0$ and so $O(X)^C \subseteq \{v \in \mathcal{V} : Y_{n+1}(v) = 0\}$. The result then follows as above. $\square$

**Remark 2.3.** *We have used the maximum over the transformed scores in order to combine score information on and off the ground truth masks. The maximum is a natural combination function in imaging and is commonly used in the context of multiple testing (Worsley et al., 1992). However the theory above is valid for any increasing combination function. We show this in Appendix A.1 where we establish generalized versions of these results.*

**Remark 2.4.** *Inner and outer coverage can also be viewed as a special case of conformal risk control with an appropriate choice of loss function. We can thus instead establish coverage results as a corollary to risk control, see Appendix A.2 for details. This amounts to an alternative proof of the results as the proof of the validity of risk control is different though still strongly relies on exchangeability. Note that in our setting risk control with the binary loss function is equivalent to the approach of Mossina et al. (2024), which when applied directly corresponds to providing the outer sets using logit scores (obtained via the identity transformation) in the applications below.*

## 2.3 JOINT CONFIDENCE SETS

Instead of focusing on marginal control one can instead spend all of the $\alpha$ available to construct sets which have a joint probabilistic guarantees. This gain comes at the expense of a loss of precision. The simplest means of constructing jointly valid confidence sets is via the marginal sets themselves.

**Corollary 2.5.** *(Joint from marginal) Assume Assumptions 1 and 2 hold and given $\alpha \in (0, 1)$ and $\alpha_1, \alpha_2 \in (0, 1)$ such that $\alpha_1 + \alpha_2 \leq \alpha$, define $I(X)$ and $O(X)$ as in Theorems 2.1 and 2.2. Then*

$$\mathbb{P}\left(I(X_{n+1}) \subseteq \{v \in \mathcal{V} : Y_{n+1}(v) = 1\} \subseteq O(X_{n+1})\right) \geq 1 - \alpha. \tag{7}$$

Alternatively joint control can be obtained using the joint distribution of the maxima of the transformed logit scores as follows.

**Theorem 2.6.** *(Joint coverage) Assume that Assumption 1 and 2 hold. Given $\alpha \in (0, 1)$, define*

$$\lambda(\alpha) = \inf\left\{\lambda : \frac{1}{n}\sum_{i=1}^{n} 1\left[\max(\tau_i, \gamma_i) \leq \lambda\right] \geq \frac{\lceil (1-\alpha)(n+1) \rceil}{n}\right\}.$$

*Let $O(X) = \{v \in \mathcal{V} : -f_O(s(X), v) \leq \lambda(\alpha)\}$ and $I(X) = \{v \in \mathcal{V} : f_I(s(X), v) > \lambda(\alpha)\}$. Then,*

$$\mathbb{P}\left(I(X_{n+1}) \subseteq \{v \in \mathcal{V} : Y_{n+1}(v) = 1\} \subseteq O(X_{n+1})\right) \geq 1 - \alpha. \tag{8}$$

*Proof.* Exchangeability of the image pairs implies exchangeability of the sequence $(\tau_i, \gamma_i)_{i=1}^{n+1}$. Moreover on the event that $\max(\tau_{n+1}, \gamma_{n+1}) \leq \lambda(\alpha)$ we have $\tau_{n+1} \leq \lambda(\alpha)$ and $\gamma_{n+1} \leq \lambda(\alpha)$ so the result follows via a proof similar to that of Theorems 2.1 and 2.2. $\qquad\square$

**Remark 2.7.** *The advantage of Corollary 2.5 is that the resulting inner and outer sets provide pivotal inference - not favouring one side or the other - which can be important when the distribution of the score function is asymmetric. Moreover the levels $\alpha_1$ and $\alpha_2$ can be used to provide a greater weight to either inner or outer sets whilst maintaining joint coverage. Theorem 2.6 may instead be useful when there is strong dependence between $\tau_{n+1}$ and $\gamma_{n+1}$. However, when this dependence is weak, scale differences in the scores can lead to a lack of pivotality. This can be improved by appropriate choices of the score transformations $f_I$ and $f_O$ however in practice it may be simpler to construct joint sets using Corollary 2.5.*

## 2.4 OPTIMIZING SCORE TRANSFORMATIONS

The choice of score transformations $f_I$ and $f_O$ is extremely important and can have a large impact on the size of the conformal confidence sets. The best choice depends on both the distribution of the data and on the nature of the output of the image segmentor used to calculate the scores. We thus recommend setting aside a learning dataset independent from both the calibration dataset, used to compute the conformal thresholds, and the test dataset. This approach was used in Sun & Yu (2024) to learn the best copula transformation for combining dependent data streams.

In order to make efficient use of the data available, the learning dataset can in fact contain some or all of the data used to train the image segmentor. This data is assumed to be independent of the calibration and test data and so can be used to learn the best score transformations without

compromising subsequent validity. The advantage of doing so is that less additional data needs to be set aside or collected for the purposes of learning a score function. Moreover it allows for additional data to be used to train the model resulting in better segmentation performance. The disadvantage is that machine learning models typically overfit their training data meaning that certain score functions may appear to perform better on this data than they do in practice. The choice of whether to include training data in the learning dataset thus depends on the quantity of data available and the quality of the segmentation model. *We do not recommend using the training data as part of the learning dataset if there is a large amount of data available as doing so may not lead to the optimal transformation.*

A score transformation that we will make particular use of in Section 3 is based on the distance transformation which we define as follows. Given $\mathcal{A} \subseteq \mathcal{V}$, let $E(\mathcal{A})$ be the set of points on the boundary of $\mathcal{A}$ obtained using the marching squares algorithm (Maple, 2003). Given a distance metric $\rho$ define the distance transformation $d_\rho : \mathcal{P}(\mathcal{V}) \times \mathcal{V} \to \mathbb{R}$, which sends $\mathcal{A} \in \mathcal{P}(\mathcal{V})$ and $v \in \mathcal{V}$ to

$$d_\rho(\mathcal{A}, v) = \text{sign}(\mathcal{A}, v) \min\{\rho(v, e) : e \in E(\mathcal{A})\},$$

where $\text{sign}(\mathcal{A}, v) = 1$ if $v \in \mathcal{A}$ and equals $-1$ otherwise. The function $d_\rho$ is an adapation of the distance transform of Borgefors (1986) which provides positive values within the set $\mathcal{A}$ and negative values outside of $\mathcal{A}$. *Moreover define the Hausdorff distance between two sets $\mathcal{A}, \mathcal{B} \subseteq \mathcal{V}$ as $H_\rho(\mathcal{A}, \mathcal{B}) = \max\{\sup_{a \in \mathcal{A}} \inf_{b \in \mathcal{B}} \rho(a, b), \sup_{b \in \mathcal{B}} \inf_{a \in \mathcal{A}} \rho(b, a)\}$, The following result shows that transforming the scores using the distance transformation ensures that accurate segmentation provides precise confidence sets. See Section A.3 for a proof.*

**Theorem 2.8.** *For each $v \in \mathcal{V}$, let $f_O(s(X), v) = d_\rho(\hat{M}(X), v)$ and define $O(X)$ as in Section 2.2. Suppose that $H_\rho(\hat{M}(X_i), Y_i) \leq k$, some $k \in \mathbb{R}$, for all $i \in J$, for some $J \subseteq \{1, \ldots, n\}$ such that $\frac{|J|}{n} > 1 - \alpha_2$. Then $H_\rho(\hat{M}(X_{n+1}), O(X_{n+1})) \leq k$. In particular if $H_\rho(\hat{M}(X_{n+1}), Y_{n+1}) \leq k$, then it follows that $H_\rho(O(X_{n+1}), Y_{n+1}) \leq 2k$.*

*A similar result holds for the inner confidence sets, see Theorem A.4. Note that a corresponding result is not true for the untransformed logit scores, see e.g. Figure A20.*

### 2.5 Constructing confidence sets from bounding boxes

Existing work on conformal inner and outer confidence sets, which aim to provide coverage of the entire ground truth mask with a given probability, has primarily focused on bounding boxes (de Grancey et al., 2022; Andéol et al., 2023; Mukama et al., 2024). These papers adjust for multiple comparisons over the 4 edges of the bounding box, doing so conformally by comparing the distance between the predicted bounding box and the bounding box of the ground truth mask. These approaches provide box-wise coverage by aggregating the predictions over all objects within all of the calibration images, often combining multiple bounding boxes per image. However, as observed in Section 5 of de Grancey et al. (2022), doing so violates exchangeability which is needed for valid conformal inference, as there is dependence between the objects within each image. Instead image-wise coverage can be provided without violating exchangeability by treating the union of the boxes as the ground truth image (de Grancey et al., 2022; Andéol et al., 2023).

We establish the validity of a version of the image-wise max-additive method of Andéol et al. (2023) (adapted to provide coverage of the ground truth) as a corollary to our results, see Appendix A.4. In this approach we define bounding box scores based on the chessboard distance transformation to the inner and outer predicted masks and use these scores to provide conformal confidence sets. Validity then follows as a consequence of the results above as we show in Corollaries A.6 and A.7. *Using the bounding box scores thus provides the same confidence sets as those used in Andéol et al. (2023).* We compare to this approach in our experiments below. Targeting bounding boxes does not directly target the mask itself and so the resulting confidence sets are typically conservative.

## 3 Application to polyps segmentation

In order to illustrate and validate our approach we consider the problem of polyps segmentation. To do so we use the same dataset as in Angelopoulos et al. (2024). The resulting dataset consists of 1798 polyps images *(from different patients)*, with available ground truth masks which were combined from 5 open-source datasets (Pogorelov et al. (2017), Borgli et al. (2020) Bernal et al.

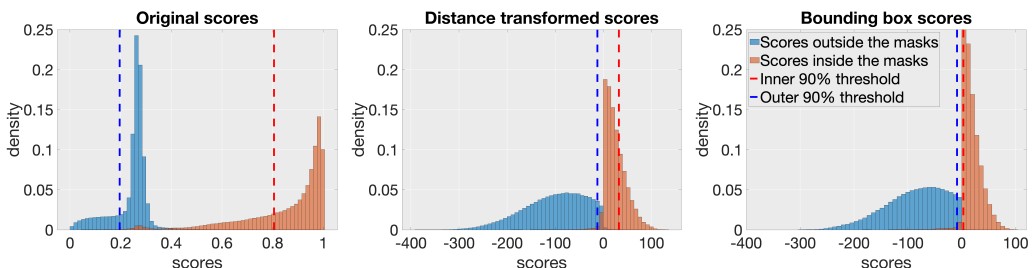

Figure 1: Histograms of the distribution of the scores over the whole image within and outside the ground truth masks. Thresholds obtained for the marginal 90% inner and outer confidence sets, obtained based on quantiles of the distribution of $(\tau_i)_{i=1}^n$ and $(\gamma_i)_{i=1}^n$, are displayed in red and blue.

(2012), Silva et al. (2014)). Logit scores were obtained for these images using the parallel reverse attention network (PraNet) model (Fan et al., 2020).

### 3.1 Choosing a score transformation

In order to optimize the size of our confidence sets we set aside 298 of the 1798 polyps images to form a learning dataset on which to choose the best score transformations. Importantly as the learning dataset is independent of the *1500 images which we set aside*, we can study it as much as we like without compromising the validity of the follow-up analyses in Sections 3.2. In particular in this section we shall use the learning dataset to both calibrate and study the results, in order to maximize the amount of important information we can learn from it.

The score transformations we considered were the identity (after softmax transformation) and distance transformations of the predicted masks: taking $f_I(s(X), v) = f_O(s(X), v) = d_\rho(\hat{M}(X), v)$, where $\rho$ is the Euclidean metric. We also compare to the results of using the bounding box transformations $f_I = b_I$ and $f_O = b_O$ which correspond to transforming the predicted bounding box using a distance transformation based on the chessboard metric and are defined formally in Appendix A.4. For the purposes of plotting we used the combined bounding box scores defined in Definition A.5.

From the histograms in Figure 1 we can see that thresholding the logit scores at the inner threshold well separates the data. However this is not the case for the outer threshold for which the data is better separated using the distance transformed and bounding box scores. Figure 2 shows PraNet scores for 2 typical examples, along with surface plots of the transformed scores and corresponding 90% marginal confidence regions (with thresholds obtained from calibrating over the learning dataset). From these we see that PraNet typically assigns a high score to the polyps regions which decreases in the regions directly around the boundary before returning to a higher level away from the polyps. This results in tight inner sets but large outer sets as the model struggles to identify where the polyps ends. Instead the distance transformed and bounding box scores are much better at providing outer bounds on the polyps, with distance transformed scores providing a tighter outside fit. Additional examples are shown in Figures A8 and A9 and have the same conclusion.

Based on the results of the learning dataset we decided to combine the best of the approaches for the inner and outer sets respectively for the inference in Section 3.2, taking $f_I$ to be the identity and $f_O$ to be the distance transformation of the predicted mask in order to optimize performance. We can also use the learning dataset to determine how to weight the $\alpha$ used to obtain joint confidence sets. A ratio of 4 to 1 seems appropriate here in light of the fact that in this dataset identifying where a given polyps ends appears to be more challenging than identifying pixels where we are sure that there is a polyps. To achieve joint coverage of 90% this involves taking $\alpha_1 = 0.02$ and $\alpha_2 = 0.08$.

### 3.2 Illustrating the performance of conformal confidence sets

In order to illustrate the full extent of our methods in practice we divide the *remaining 1500* images at random into 1000 for conformal calibration, and 500 for testing. The resulting conformal confidence sets for 10 example images from the test dataset are shown in Figure 3, with inner sets obtained using the untransformed logit scores and outer sets using the distance transformed scores.

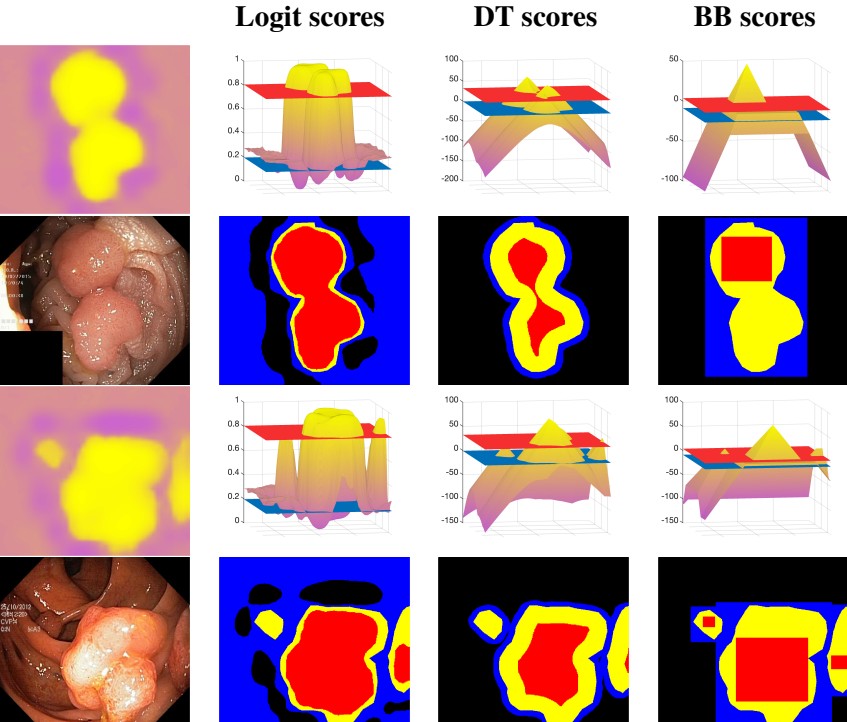

Figure 2: Illustrating the performance of the different score transformations on the learning dataset. We display 2 example polyps images and present the results of each in 8 panels. These panels are as follows. Bottom left: the original image of the polyps. Top Left: an intensity plot of the scores obtained from PraNet with purple/yellow indicating areas of lower/higher assigned probability. For the remaining panels, 3 different score transformations are shown which from left to right are the untransformed logit scores, distance transformed (DT) scores $d_\rho(\hat{M}(X), v)$ and bounding box (BB) scores (obtained using the combined bounding box score $b_M$ defined in Definition A.5). In each of the panels on the top row a surface plot of the transformed PraNet scores is shown, along with the conformal thresholds which are used to obtain the marginal 90% inner and outer confidence sets. These thresholds are illustrated via red and blue planes respectively and are obtained over the learning dataset. The panels on the bottom row of each example show the corresponding conformal confidence sets. Here the inner set is shown in red, plotted over the ground truth mask of the polyps, shown in yellow, plotted over the outer set which is shown in blue. The outer set contains the ground truth mask which contains the inner set in all examples. From these figures we see that the logit scores provide tight inner confidence sets and the distance transformed scores instead provide tight outer confidence sets. The conclusion from the learning dataset is therefore that it makes sense to combine these two score transformations.

*Details of the test time implementation are shown in Algorithm 1*. The inner sets are shown in red and represent regions where we can have high confidence of the presence of polyps. The outer sets are shown in blue and represent regions in which the polyps may be. The ground truth mask for each polyps is shown in yellow and can be compared to the original images. In each of the examples considered the ground truth is bounded from within by the inner set and from without by the outer set. Results for confidence sets based on the logit and bounding box scores as well as additional examples are available in Figures A10 and A11. Confidence sets can also be provided for the bounding boxes themselves if that is the object of interest, see Figure A12. Joint 90% confidence sets are displayed in Figure A13, from which we can see that with alpha-weighting (i.e. taking $\alpha_1 = 0.02$ and $\alpha_2 = 0.08$) we are able to obtain joint confidence sets which are still relatively tight.

These results collectively show that we can provide informative confidence bounds for the location of the polyps and allow us to use the PraNet segmentation model with uncertainty guarantees. From Figure 3 we can see that the method, which combines the logit and the distance transformed scores, effectively delineates polyps regions. These results also help to make us aware of the limitations of the model, allowing medical practioners to follow up on outer sets which do not contain inner sets in

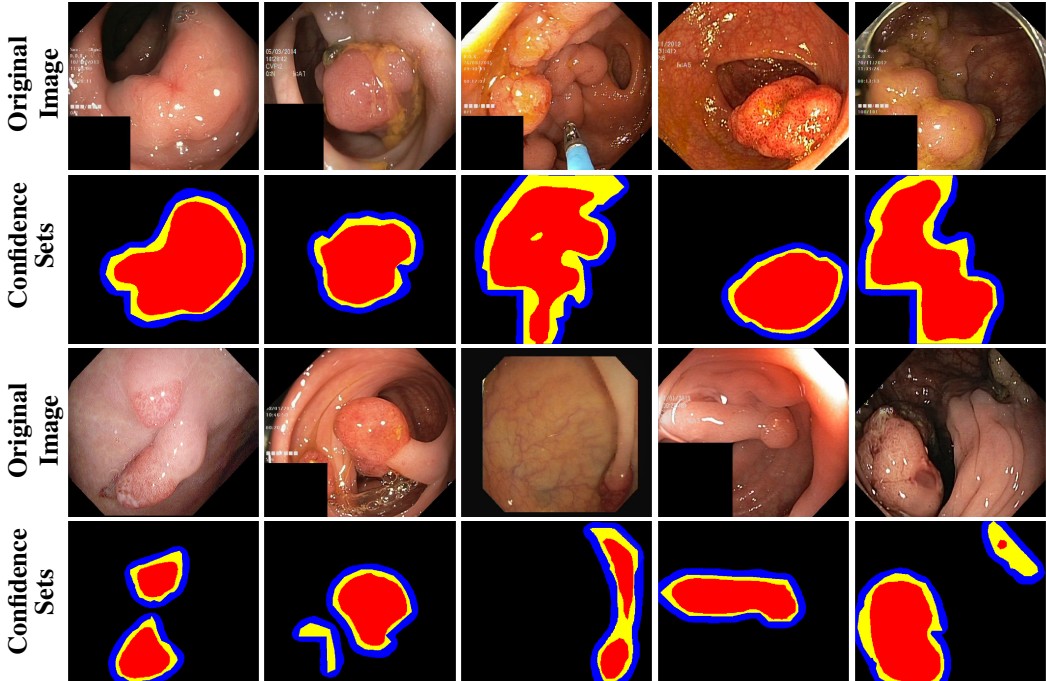

Figure 3: Conformal confidence sets for the polyps data. For each set of polyps images the top row shows the original endoscopic images with visible polyps and the second row presents the marginal 90% confidence sets, with ground truth masks shown in yellow. The inner sets and outer sets are shown in red and blue, obtained using the identity and distance transforms respectively. The figure shows the benefits of combining different score transformations for the inner and outer sets and illustrates the method's effectiveness in accurately identifying polyp regions whilst providing informative spatial uncertainty bounds.

order to determine whether a polyps is present. Improved uncertainty quantification would require an improved segmentation model.

More precise results can be obtained at the expense of probabilistic guarantees, see Figures A14 and A15. A trade off must be made between precision and confidence. The most informative confidence level can be determined in advance based on the learning dataset and the desired type of coverage.

### 3.3 MEASURING THE COVERGE RATE

In this section we run validations to evaluate the false coverage rate of our approach. To do so we take the *1500 images which we set aside* and run 1000 validations, in each validation dividing the data into 1000 calibration and 500 test images. In each division we calculate the conformal confidence sets using the different score transformations, based on thresholds derived from the calibration dataset, and evaluate the coverage rate on the test dataset. We average over all 1000 validations and present the results in Figure 4. Histograms for the 90% coverage obtained over all validation runs are shown in Figure A16.

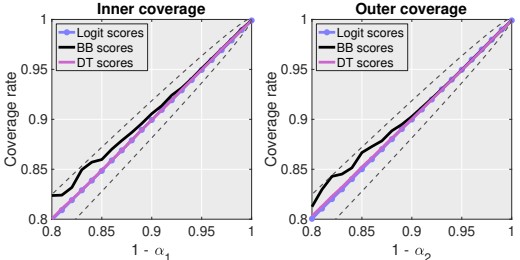

Figure 4: Coverage levels of the inner and outer sets averaged over 1000 validations for the logit, distance transformed (DT) and bounding box (BB) scores. *95% uncertainty bands are shown with the dashed grey lines.*

From these results we can see that for all the approaches the coverage rate is controlled at or above the nominal level as desired. Using the bounding box scores results in slight over coverage at lower confidence levels. This is likely due to the discontinuities in the score functions $b_I$ and $b_O$.

## 3.4 COMPARING THE EFFICIENCY OF THE BOUNDS

In this section we compare the efficiency of the confidence sets based on the different score transformations. To do so we run 1000 validations in each dividing and calibrating as in Section 3.3.

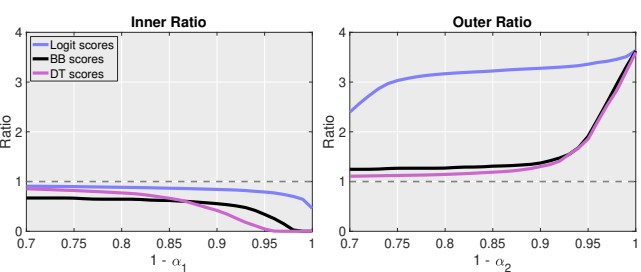

For each run we compute the ratio between the diameter of the inner set and the diameter of the ground truth mask and average this ratio over the 500 test images. In order to make a smooth curve we average this quantity over all 1000 runs. A similar calculation is performed for the outer set. The results are shown in Figure 5. They show that the inner confidence sets produced by using the logit scores are the most efficient. Instead, for the outer set, the distance transformed scores perform best. These results match the observations of Sections 3.1 and 3.2.

Figure 5: Measuring the efficiency of the bound using the ratio of the diameter of the coverage set to the diameter of the true mask. The closer the ratio is to one the better. Higher coverage rates lead to a lower efficiency. The logit scores provide the most efficient inner sets and the distance transformed scores provide the most efficient outer sets.

We repeat this procedure instead targeting the proportion of the entire image which is under/over covered by the respective confidence sets. The results are shown in Figure A17 and can be interpreted similarly.

## 4 APPLICATION TO BRAIN IMAGING SEGMENTATION

*As a second application we consider the task of skull stripping. This task consists of segmenting the brain given an Magnetic resonance image of a human head. For image segmentation we use the HD-BET (Isensee et al., 2019) neural network model which was trained on dataset of 1,568 subjects and has quickly become the defacto method of performing brain mask segmentation. In order to apply our methods in this setting we combine data from 3 public datasets (LPBA40, NFBS, and*

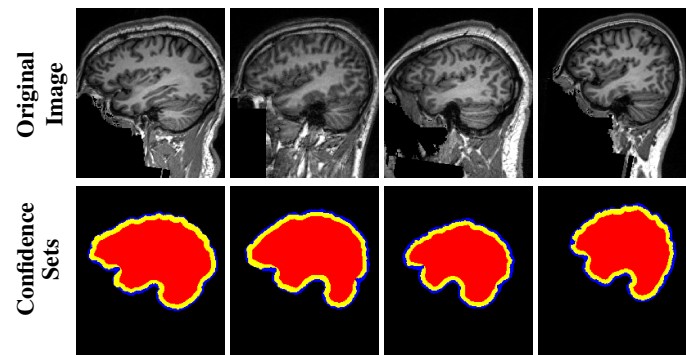

Figure 6: *Inner and outer confidence sets for brain mask segmentation: both computed using the distance transformed scores. The true mask is shown in yellow.*

*CC-359) resulting in 524 brain images in total. This data is independent from the data used to train HD-BET, see e.g. (Isensee et al., 2019). We divide this data into 50 subjects to make up a learning dataset, use 300 subjects to perform calibration and use the remaining subjects for testing.*

*Based on the results of the learning dataset, see Appendix A.7.1, we see that the distance transformed scores perform best for constructing both inner and outer confidence sets. The naive approach of using the untransformed logit scores performs very poorly for both inner and outer confidence sets, see Figure A18. Calibrating thresholds for the distance transformed scores using the calibration dataset and applying to the images from the testing dataset we instead obtain informative inner and outer confidence sets, as shown in Figure 6, see also Figure A20. Validating as in Section 3.3, we see that the false coverage rate is controlled to the nominal level, see Section A.7.4.*

## 5 APPLICATION TO TEETH SEGMENTATION

*As a third application we consider the problem of teeth segmentation. We use a dataset (released by Zhang et al. (2023)) consisting of scans of the teeth of 598 subjects and train a U-net based GAN network using 400 subjects (following Hoshme (2024)). We divide the remaining 198 subjects into 170 to use as calibration data and 28 to use as a test dataset. We use the original training data as a*

*learning dataset (note that this is independent of the calibration dataset so does not affect validity). We tried a variety of score transformations including distance transformations and smoothing, see Section A.8. Based on the learning data we chose the distance transformation for the outer sets. For the inner set we instead use scores smoothed with an isotropic Gaussian kernel with full width at half maximum (FWHM) of 2 pixels. Calibrating thresholds on the calibration dataset and applying to the test data we obtain the results shown in Figure 7. Moreover, validating as in Section 3.3, we show that the false coverage rate is controlled to the nominal level in practice, see Section A.8.4.*

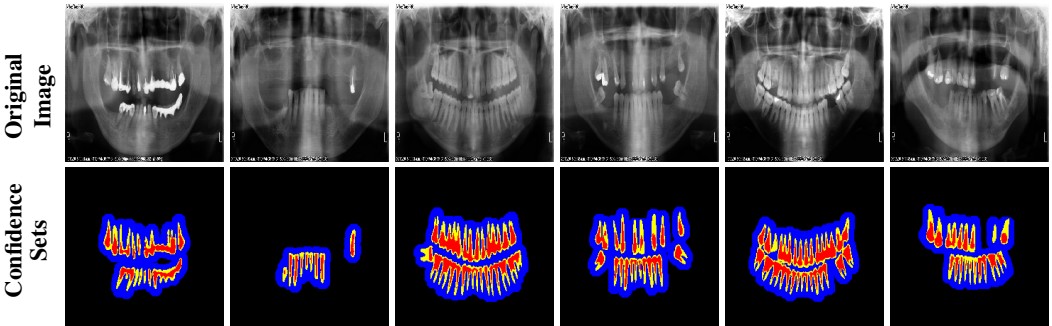

Figure 7: *Inner and outer confidence sets for teeth segmentation computed using scores smoothed with 2 pixel FWHM and distance transformed scores respectively. The true mask is shown in yellow.*

## 6 DISCUSSION

In this work, we have developed conformal confidence sets which offer probabilistic guarantees for the output of a black box image segmentation model and provide tight bounds. Our work helps to address the lack of formal uncertainty quantification in the application of deep neural networks to medical imaging which has limited the reliability and adoption of these models in practice. *Confidence sets provide informative spatial bounds on the expected output and ensure that we are not overconfident about our model predictions.*

The use of the distance transformed scores *was important in providing tight outer confidence bounds in all applications considered* as the original neural network is by itself unable to reliably determine where the true masks end with certainty. The distance transformation penalizes regions away from the predicted mask, allowing the true mask to be distinguished from the background. In other datasets and model settings, other transformations may be appropriate. We saw for instance that smoothing the scores can be beneficial and allow the model to boost power using spatial information. As such we strongly recommend the use of a learning dataset to learn the best transformation and maximize the precision of the resulting confidence bounds.

*We have shown (Theorems 2.8 and A.4) that an increase in the quality of the predictions of the image segmentation model leads to more precise confidence sets when using the distance transformed scores. Such a relationship does not hold for the untransformed logit scores. This is well illustrated in the brain imaging application, see Figure A20, in which the distance transformed scores allow for very tight uncertainty bands but the confidence sets obtained based on the logit scores are very uninformative. Metrics for each model used are shown in Appendix A.9 and the model (HDBET) with the highest performance on these metrics has the most precise confidence bands. Further refinements to the segmentations within these bounds could be acheived using biological information on the shapes of the ground truth masks. In this work we have assumed that a ground truth mask is known for each subject. However, in some applications mutliple raters provide true masks for each segmentation. Future work could look at incorporating the uncertainty in the rating process into the conformal prediction algorithm to better infer on model uncertainties.*

Matlab code to implement the methods of this paper and a demo on a downscaled version of the data is available in the supplementary material. The code is very fast: calculating inner and outer thresholds (over the 1000 images in the calibration set) requires approximately 0.03 seconds on the downscaled polyps data on a standard laptop (Apple M3 chip with 16 GB RAM) and 2.64 seconds for the original polyps dataset.

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

# A APPENDIX

## A.1 OBTAINING CONFORMAL CONFIDENCE SETS WITH INCREASING COMBINATION FUNCTIONS

As discussed in Remark 2.3 the results of Sections 2.2 and 2.3 can be generalized to a wider class of combination functions.

**Definition A.1.** We define a suitable combination function to be a function $C : \mathcal{P}(\mathcal{V}) \times \mathcal{X} \to \mathbb{R}$ which is increasing in the sense that for all sets $\mathcal{A} \subseteq \mathcal{V}$ and each $v \in \mathcal{A}$, $C(v, X) \leq C(\mathcal{A}, X)$ for all $X \in \mathcal{X}$.

The maximum is a suitable combination function since $X(v) = \max_{v \in \{v\}} X(v) \leq \max_{v \in \mathcal{A}} X(v)$. As such this framework directly generalizes the results of the main text.

We can construct generalized marginal confidence sets as follows.

**Theorem A.2.** *(Marginal inner set) Under Assumptions 1 and 2, given $\alpha_1 \in (0, 1)$, define*

$$\lambda_I(\alpha_1) = \inf \left\{ \lambda : \frac{1}{n} \sum_{i=1}^{n} 1 \left[ C(\{v \in \mathcal{V} : Y_i(v) = 1\}, f_I(s(X_i))) \leq \lambda \right] \geq \frac{\lceil (1 - \alpha_1)(n + 1) \rceil}{n} \right\},$$

*for a suitable combination function $C$, and define $I(X) = \{v \in \mathcal{V} : C(v, f_I(s(X))) > \lambda_I(\alpha_1)\}$. Then,*

$$\mathbb{P}\left(I(X_{n+1}) \subseteq \{v \in \mathcal{V} : Y_{n+1}(v) = 1\}\right) \geq 1 - \alpha_1. \tag{9}$$

The proof follows that of Theorem 2.1. The key observation is that for any suitable combination function $C$, given $\lambda \in \mathbb{R}$, $\mathcal{A} \subseteq \mathcal{V}$ and $X \in \mathcal{X}$, $C(\mathcal{A}, X) \leq \lambda$ implies that $C(v, X) \leq \lambda$. This is the relevant property of the maximum which we used for the results in the main text. For the outer set we similarly have the following.

**Theorem A.3.** *(Marginal outer set) Under Assumptions 1 and 2, given $\alpha_2 \in (0, 1)$, define*

$$\lambda_O(\alpha_2) = \inf \left\{ \lambda : \frac{1}{n} \sum_{i=1}^{n} 1 \left[ C(\{v \in \mathcal{V} : Y_i(v) = 0\}, -f_O(s(X_i))) \leq \lambda \right] \geq \frac{\lceil (1 - \alpha_2)(n + 1) \rceil}{n} \right\}.$$

*for a suitable combination function $C$, and let $O(X) = \{v \in \mathcal{V} : C(v, -f_O(s(X))) \leq \lambda_O(\alpha_2)\}$. Then,*

$$\mathbb{P}\left(\{v \in \mathcal{V} : Y_{n+1}(v) = 1\} \subseteq O(X_{n+1})\right) \geq 1 - \alpha_2. \tag{10}$$

Joint results can be analogously obtained.

## A.2 OBTAINING CONFIDENCE SETS FROM RISK CONTROL

We can alternatively establish Theorems 2.1 and A.2 using an argument from risk control (Angelopoulos et al., 2024). In particular, given an image pair $(X, Y)$ and $\lambda \in \mathbb{R}$, let

$$I_\lambda(X) = \{v \in \mathcal{V} : f_I(s(X), v) > \lambda\}.$$

Define a loss function, $L : \mathcal{P}(\mathcal{V}) \times \mathcal{Y} \to \mathbb{R}$ which sends $(X, Y)$ to

$$L(I_\lambda(X), Y) = 1 \left[ I_\lambda(X) \not\subseteq \{v \in \mathcal{V} : Y(v) = 1\} \right].$$

For $i = 1, \ldots, n + 1$, let $L_i(\lambda) = L(I_\lambda(X_i), Y_i)$. Arguing as in the proof of Theorem 2.1 it follows that $L_i(\lambda) = 1[\tau_i > \lambda]$. Then applying Theorem 1 of Angelopoulos et al. (2024) it follows that

$$\mathbb{E}\left[L_{n+1}(\hat{\lambda})\right] \leq \alpha_1,$$

where $\hat{\lambda} = \inf \left\{ \lambda : \frac{1}{n} \sum_{i=1}^{n} L_i(\lambda) \leq \alpha_1 - \frac{1 - \alpha_1}{n} \right\}$. Arguing as in Appendix A of (Angelopoulos et al., 2024) it follows that

$$\hat{\lambda} = \inf \left\{ \lambda : \frac{1}{n} \sum_{i=1}^{n} 1 \left[ \tau_i \leq \lambda \right] \geq \frac{\lceil (1 - \alpha_1)(n + 1) \rceil}{n} \right\} = \lambda_I(\alpha_1),$$

and so $I(X) = I_{\hat{\lambda}}(X)$. As such

$$\mathbb{P}\left(I(X_{n+1}) \subseteq \{v \in \mathcal{V} : Y_{n+1}(v) = 1\}\right) = 1 - \mathbb{E}\left[L_{n+1}(\hat{\lambda})\right] \geq 1 - \alpha_1, \tag{11}$$

and we recover the desired result. Arguing similarly it is possible to establish a proof of Theorem 2.2.

## A.3 Characterizing the relationship between Haussdorff distance and the distance transformed scores

*In this section we provide a proof of Theorem 2.8 and state the analogous result for the inner confidence sets obtained using the distance transformation.*

*Proof.* Consider the outer confidence sets obtained using the distance transformed scores. Then given $1 \le i \le n$ such that $H(\hat{M}(X_i), Y_i) \le k$, we have

$$Y_i = \left( Y_i \cap \hat{M}(X_i) \right) \cup \left( Y_i \cap \hat{M}(X_i)^C \right)$$

where the union is disjoint. The distance transformed scores $d_\rho(\hat{M}(X_i), v)$ are positive for $v \in \hat{M}(X_i)$ and negative for $v \notin \hat{M}(X_i)$. As such

$$\gamma_i = \max_{v \in \mathcal{V}: Y_i(v)=1} -f_O(s(X_i), v) = \max_{v \in \mathcal{V}: Y_i(v)=1} -d_\rho(\hat{M}(X_i), v)$$

$$= \max_{v \in \mathcal{V}: Y_i(v)=1} \min_{e \in E(\hat{M}(X_i))} \rho(v, e) \le H_\rho(\hat{M}(X_i), Y_i) \le k.$$

Since this holds for all $i$ on a set $J$ which has $\frac{|J|}{n} > 1 - \alpha_2$, it follows that $\lambda_O(\alpha_2) \le k$. Arguing similarly in the opposite direction it follows that for any new observation $X_{n+1}$ we have that

$$H_\rho(\hat{M}(X_{n+1}), O(X_{n+1})) \le k.$$

Finally if $H_\rho(\hat{M}(X_{n+1}), Y_{n+1}) \le k$, then it follows that $H_\rho(O(X_{n+1}), Y_{n+1}) \le 2k$ by the triangle inequality. □

*A similar result can be established for the inner confidence sets via an analogous proof. We state this formally as follows.*

**Theorem A.4.** *For each $v \in \mathcal{V}$, let $f_I(s(X), v) = d_\rho(\hat{M}(X), v)$ and define $I(X)$ as in Section 2.2. Suppose that $H_\rho(\hat{M}(X_i), Y_i) \le k$, some $k \in \mathbb{R}$, for all $i \in J$, for some $J \subseteq \{1, \ldots, n\}$ such that $\frac{|J|}{n} > 1 - \alpha_1$. Then $H_\rho(\hat{M}(X_{n+1}), I(X_{n+1})) \le k$. In particular if $H_\rho(\hat{M}(X_{n+1}), Y_{n+1}) \le k$, then $H_\rho(I(X_{n+1}), Y_{n+1}) \le 2k$.*

## A.4 Deriving confidence sets from bounding boxes

We can use our results in order to provide valid inference for bounding boxes via an adaption of the approach of Andéol et al. (2023). In particular given $Z \in \mathcal{Y}$, let $B_{I,\max}(Z)$ be the largest box which can be contained within the set $\{v \in \mathcal{V} : Z(v) = 1\}$ and let $B_{O,\min}(Z)$ be the smallest box which contains the set $\{v \in \mathcal{V} : Z(v) = 1\}$. Given $Y \in \mathcal{Y}$, let $cc(Y) \subseteq \mathcal{P}(\mathcal{V})$ denote the set of connected components of the set $\{v \in \mathcal{V} : Y(v) = 1\}$ for a given connectivity criterion (which we take to be 4 in our examples), and note that these components can themselves be identifed as elements of $\mathcal{Y}$. Define

$$B_I(Y) = \cup_{c \in cc(Y)} B_{I,\max}(c) \text{ and } B_O(Y) = \cup_{c \in cc(Y)} B_{O,\min}(c)$$

to be the unions of the largest inner and smallest outer boxes of the connected components of the image $Y$, respectively. Then define

$$\hat{B}_I(s(X)) = \cup_{c \in cc(\hat{M}(X))} B_{I,\max}(c) \text{ and } \hat{B}_O(s(X)) = \cup_{c \in cc(\hat{M}(X))} B_{O,\min}(c)$$

to be the unions of the largest inner and smallest outer boxes of the connected components of the predicted mask $\hat{M}(X)$, respectively. Note that this is well-defined as $\hat{M}(X)$ is a function of $s(X)$.

For the remainder of this section we shall assume that $\mathcal{V} \subset \mathbb{R}^2$, this is not strictly necessary but will help to simplify notation. Given $u, v \in \mathcal{V}$, write $u = (u_1, u_2)$ and $v = (v_1, v_2)$ and let $\rho(u, v) = \max(|u_1 - v_1|, |u_2 - v_2|)$ be the chessboard metric.

**Definition A.5.** (Bounding box scores) For each $X \in \mathcal{X}$ and $v \in \mathcal{V}$, let

$$b_I(s(X), v) = d_\rho(\hat{B}_I(s(X)), v) \text{ and } b_O(s(X), v) = d_\rho(\hat{B}_O(s(X)), v)$$

be the distance transformed scores based on the chessboard distance to the predicted inner and outer box collections $\hat{B}_I(s(X))$ and $\hat{B}_O(s(X))$, respectively. We also define a combination of these $b_M$, primarily for the purposes of plotting in Figure 2, as follows. Let $b_M(s(X), v) = b_O(s(X), v)$ for each $v \notin \hat{B}_O(s(X))$ and let $b_M(s(X), v) = \max(b_I(s(X), v), 0)$ for $v \in \hat{B}_O(s(X))$. We shall write $b_I(s(X)) \in \mathcal{X}$ to denote the image which has $b_I(s(X))(v) = b_I(s(X), v)$ and similarly for $b_O(s(X))$ and $b_M(s(X))$.

Now consider the sequences of image pairs $(X_i, B_i^I)_{i=1}^n$ and $(X_i, B_i^O)_{i=1}^n$. These both satisfy exchangeability and so, applying Theorems A.2 and A.3, we obtain the following bounding box validity results.

**Corollary A.6.** *(Marginal inner bounding boxes) Suppose Assumption 1 holds and that $(X_i, Y_i)_{i=1}^{n+1}$ is independent of the functions $s$ and $b_I$. Given $\alpha_1 \in (0, 1)$, define*

$$\lambda_I(\alpha_1) = \inf \left\{ \lambda : \frac{1}{n} \sum_{i=1}^n 1\left[ C(B_i^I, b_I(s(X_i))) \leq \lambda \right] \geq \frac{\lceil (1 - \alpha_1)(n+1) \rceil}{n} \right\}, \quad (12)$$

*for a suitable combination function $C$, and define $I(X) = \{v \in \mathcal{V} : C(v, b_I(s(X))) > \lambda_I(\alpha_1)\}$. Then,*

$$\mathbb{P}\left( I(X_{n+1}) \subseteq B_{n+1}^I \subseteq \{v \in \mathcal{V} : Y_{n+1}(v) = 1\} \right) \geq 1 - \alpha_1.$$

**Corollary A.7.** *(Marginal outer bounding boxes) Suppose Assumption 1 holds and that $(X_i, Y_i)_{i=1}^{n+1}$ is independent of the functions $s$ and $b_O$. Given $\alpha_2 \in (0, 1)$, define*

$$\lambda_O(\alpha_2) = \inf \left\{ \lambda : \frac{1}{n} \sum_{i=1}^n 1\left[ C(B_i^O, -b_O(s(X_i))) \leq \lambda \right] \geq \frac{\lceil (1 - \alpha_2)(n+1) \rceil}{n} \right\}. \quad (13)$$

*for a suitable combination function $C$, and let $O(X) = \{v \in \mathcal{V} : C(v, -b_O(s(X))) \leq \lambda_O(\alpha_2)\}$. Then,*

$$\mathbb{P}\left( \{v \in \mathcal{V} : Y_{n+1}(v) = 1\} \subseteq B_{n+1}^O \subseteq O(X_{n+1}) \right) \geq 1 - \alpha_2.$$

Joint results can be obtained in a similar manner to those in Section 2.3.

## A.5 WRITING THE TEST TIME STEPS AS AN ALGORITHM

*In order to clarify what is done at test time we include the following algorithm which demonstrates this for the polyps data application.*

---
**Algorithm 1** Test time application of the methods (for the polyps data application)

---
**Require:** Inner and outer alpha levels $\alpha_1$ and $\alpha_2$ and thresholds $\lambda_I(\alpha_1)$ and $\lambda_O(\alpha_2)$ obtained as in equations (3) and (5) from the calibration dataset with $f_I$ the identity and $f_O$ the distance transformed scores. A test time observation $X_{n+1}$ and a score function $s(X_{n+1})$ obtained by an image segmenting model and a distance metric $\rho$.

1: Compute the predicted mask as $\hat{M}(X_{n+1}) = \{v \in \mathcal{V} : s(X_{n+1}, v) > 0\}$
2: Calculate the set of points $E(\hat{M}(X_{n+1}))$ on the boundary of the predicted mask $\hat{M}(X_{n+1})$ using the marching squares algorithm.
3: Compute $d_\rho(\hat{M}(X_{n+1}), v) = \text{sign}(\hat{M}(X_{n+1}), v) \min\{\rho(v, e) : e \in E(\hat{M}(X_{n+1}))\}$, i.e. the distance transformed scores, for each $v \in \mathcal{V}$.
4: Let $I(X_{n+1}) = \{v \in \mathcal{V} : s(X, v) > \lambda_I(\alpha_1)\}$.
5: Let $O(X_{n+1}) = \{v \in \mathcal{V} : -d_\rho(\hat{M}(X_{n+1}), v) \leq \lambda_O(\alpha_2)\}$.
6: **return** $I(X_{n+1})$ and $O(X_{n+1})$.

---

*For the brain imaging application, $\lambda_I(\alpha_1)$ and $\lambda_O(\alpha_2)$ are instead computed from the calibration dataset with both $f_I$ and $f_O$ being the distance transformed scores. Then line 4 of the algorithm at test time is replaced by: Let $I(X_{n+1}) = \{v \in \mathcal{V} : d_\rho(\hat{M}(X_{n+1}), v) > \lambda_I(\alpha_1)\}$. For the teeth segmentation problem the inner scores are analogously replaced but with scores smoothed using an isotropic Gaussian kernel with FWHM 2 pixels.*

### A.6 ADDITIONAL SETTINGS FOR POLYPS SEGMENTATION

Here we plot additional settings and examples for the polyps data application. The version of the method which uses the distance transformation to create outer confidence sets and the untransformed logit scores to create inner confidence sets will be referred to as combo.

### A.6.1 ADDITIONAL EXAMPLES FROM THE LEARNING DATASET

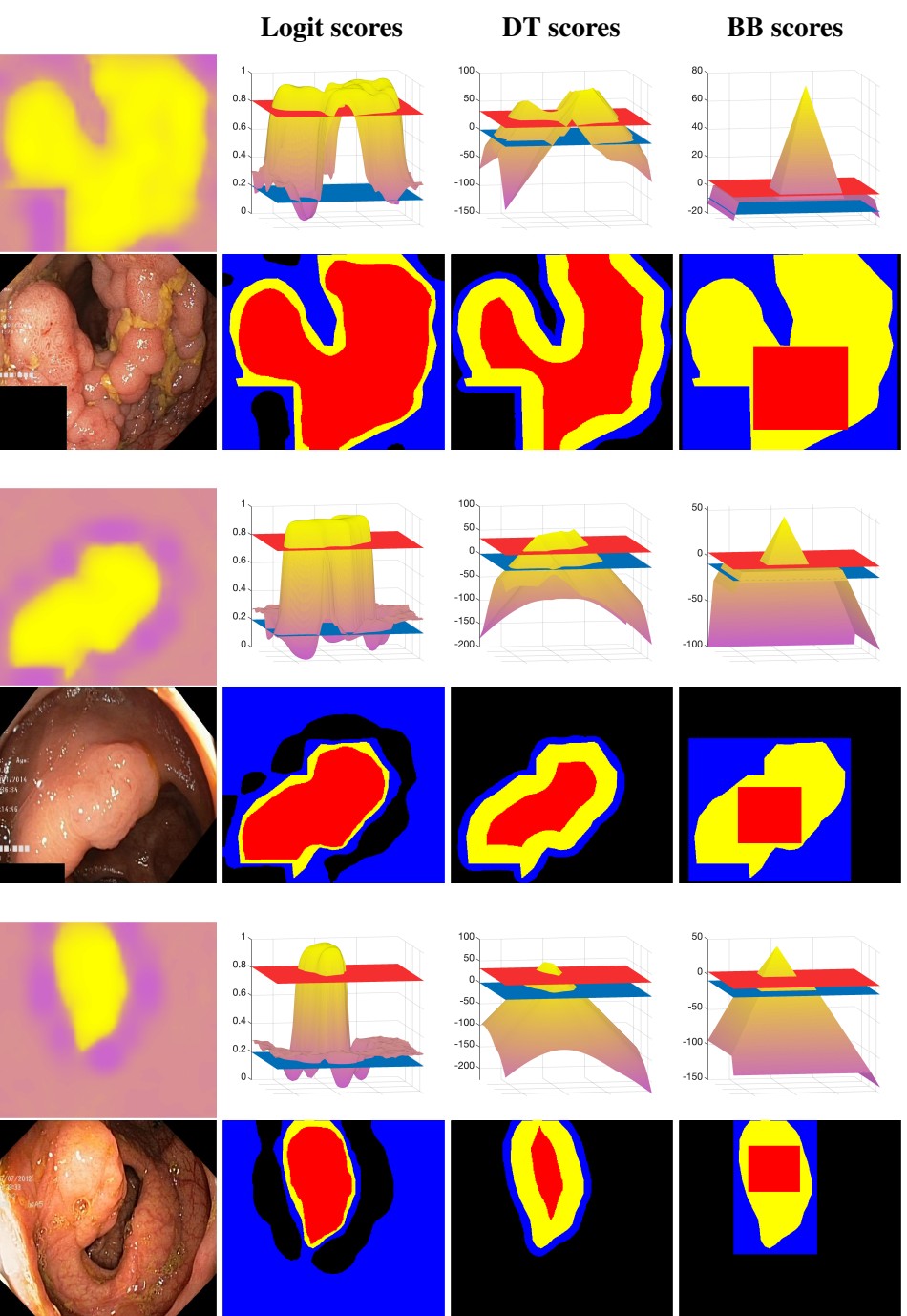

Figure A8: Additional examples from the learning dataset. The layout of these figures is the same as for Figure 2.

918
919
920
921
922
923
924
925
926
927
928
929
930
931
932
933
934
935
936
937
938
939
940
941
942
943
944
945
946
947
948
949
950
951
952
953
954
955
956
957
958
959
960
961
962
963
964
965

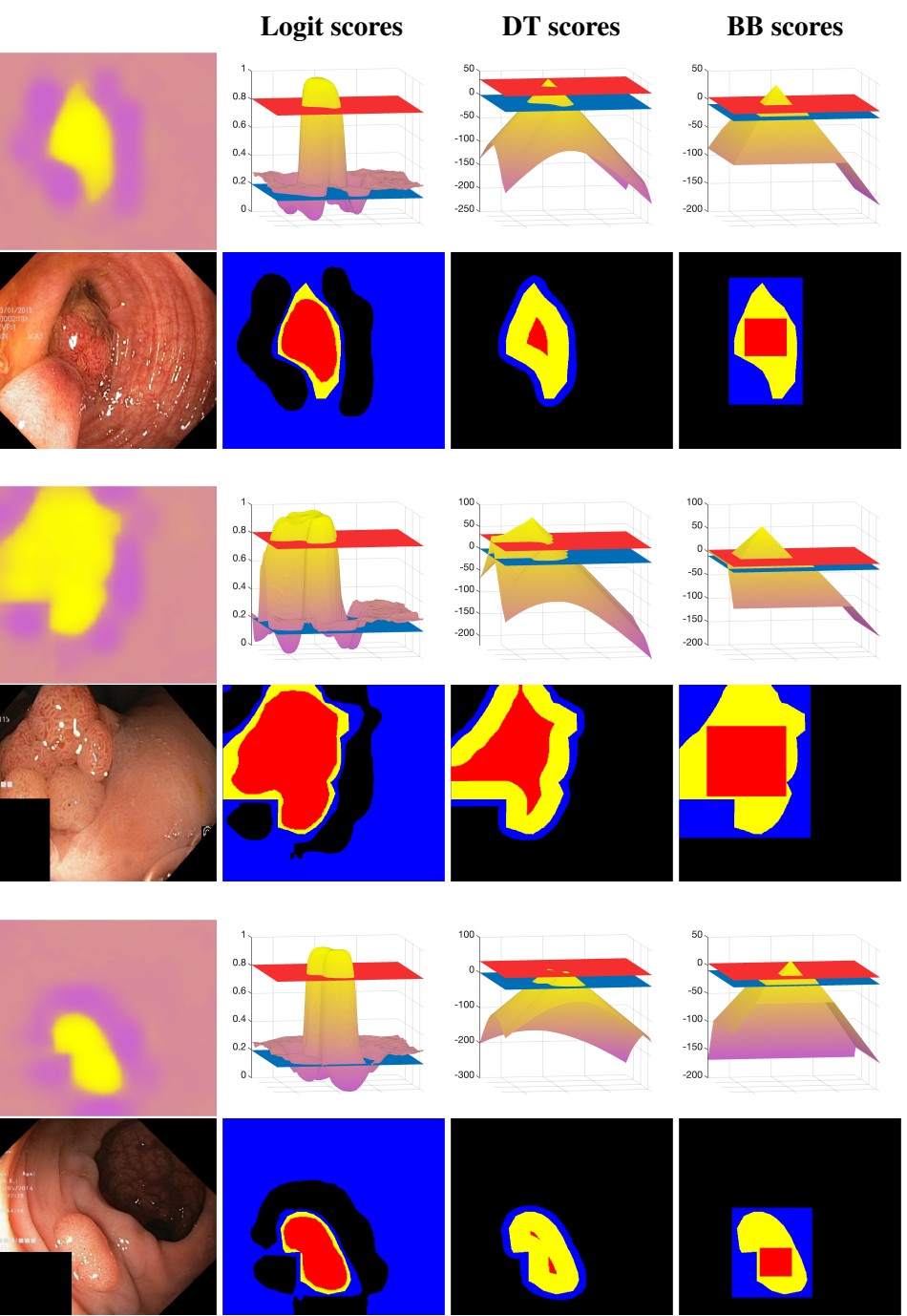

Figure A9: Futher examples from the learning dataset. The layout of these figures is the same as for Figure 2.

966
967
968
969
970
971

### A.6.2 VALIDITION FIGURES FOR THE ORIGINAL AND BOUNDING BOX SCORES

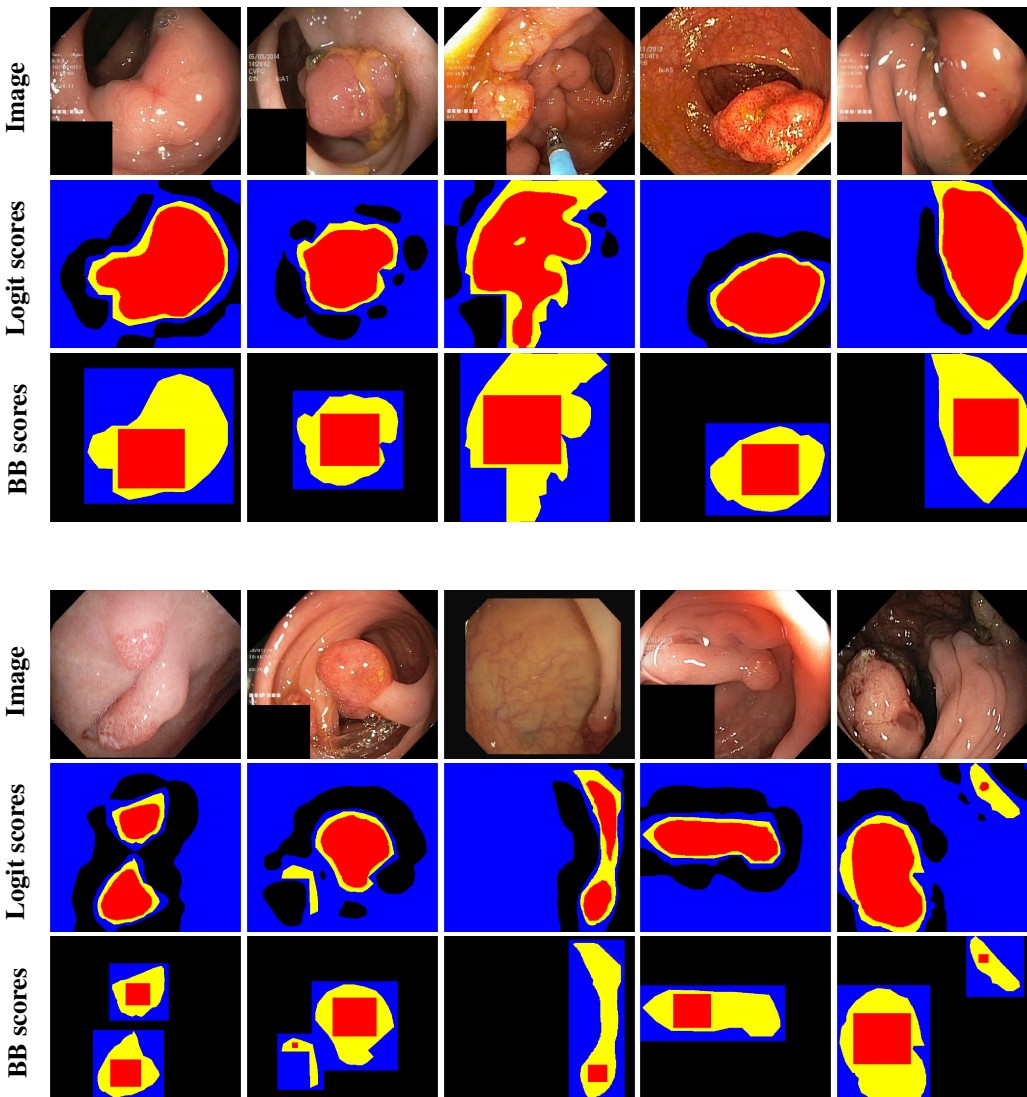

Figure A10: Conformal confidence sets for the polyps data examples from Figure 3 for alternative scores. In each set of panels the confidence obtained from using the logit scores are shown in the middle row and those obtained from the bounding box scores are shown in the bottom row. As observed on the learning dataset the outer sets obtained when using the logit scores are very large and uninformative.

### A.6.3 ADDITIONAL VALIDITION FIGURES

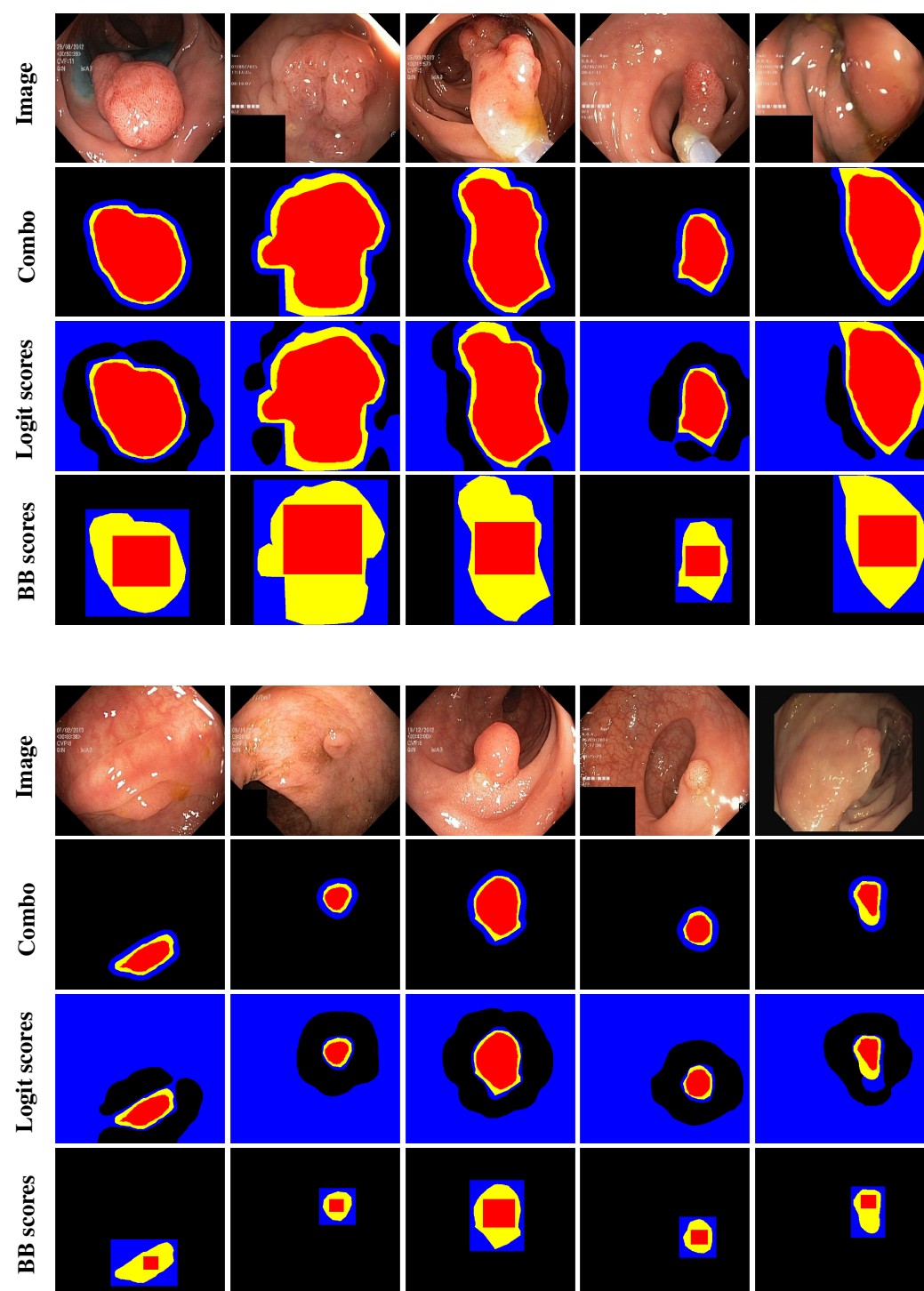

Figure A11: Additional validition examples. In each example, after the original images, the rows are (from top to bottom) the combination of the original and distance transformed scores, then the logit scores and finally the bounding box scores. The interpretation of the results is the same as for Figure 3.

### A.6.4 CONFIDENCE SETS FOR THE BOUNDING BOXES

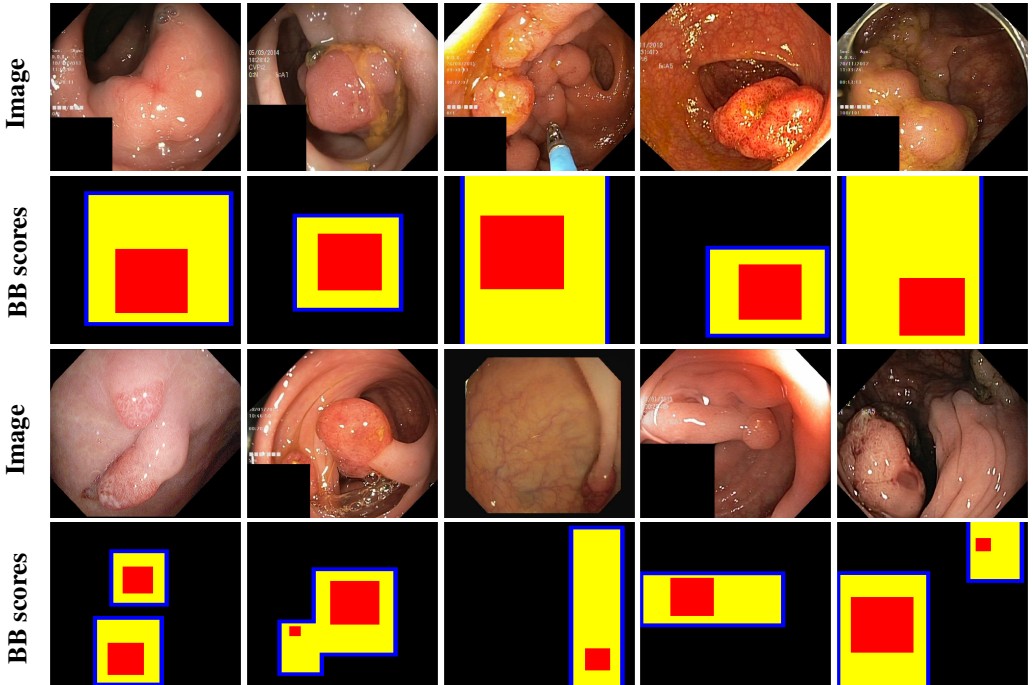

Figure A12: Conformal confidence sets for the boundary boxes themselves using the approach introduced in Section A.4. The ground truth outer bounding boxes are shown in yellow.

### A.6.5 JOINT 90% CONFIDENCE REGIONS

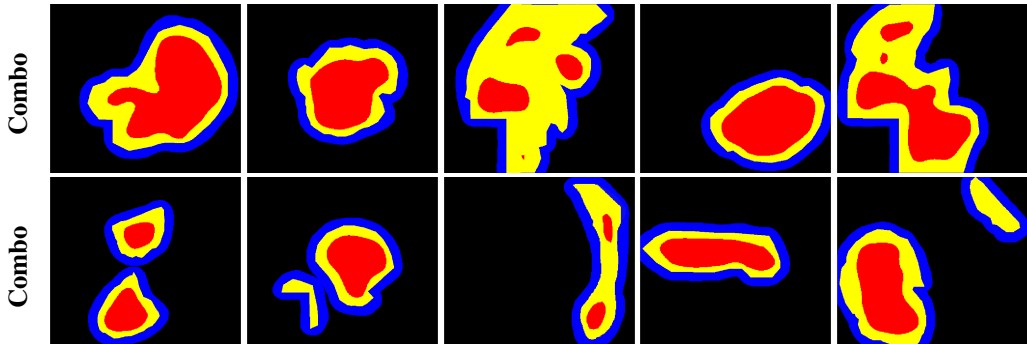

Figure A13: Joint 90% conformal confidence sets obtained using Corollary 2.5, with $\alpha_1 = 0.02$ and $\alpha_2 = 0.08$, for the polyps images in Figure 3.

### A.6.6  MARGINAL 80 % CONFIDENCE REGIONS

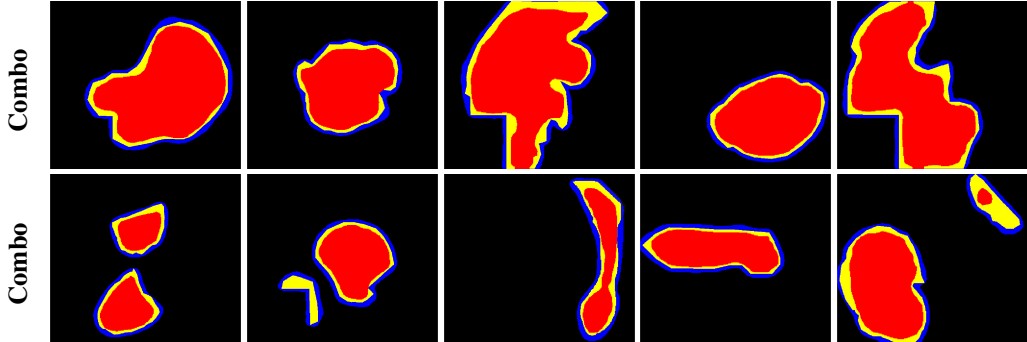

Figure A14: Marginal 80% conformal confidence sets obtained for the polyps images in Figure 3.

### A.6.7  MARGINAL 95 % CONFIDENCE REGIONS

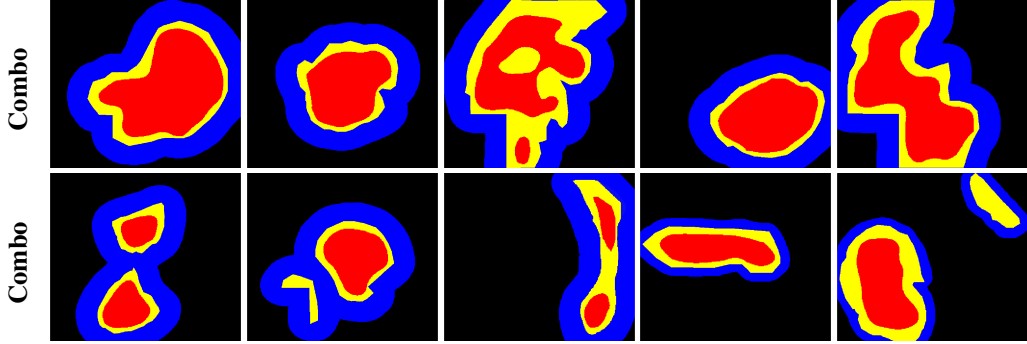

Figure A15: Marginal 95% conformal confidence sets obtained using for the polyps images in Figure 3. These sets are also joint 90% confidence sets with equally weighted $\alpha_1 = \alpha_2 = 0.05$. The influence of the weighting scheme can therefore examined by comparing to Figure A13.

### A.6.8 HISTOGRAMS OF THE COVERAGE

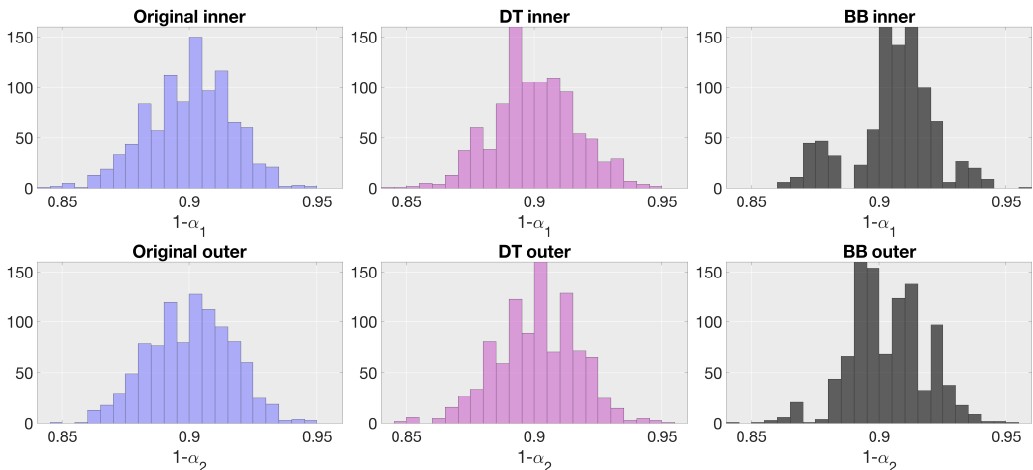

Figure A16: Histograms of the coverage rates obtained across each of the validation resamples for 90% inner and outer marginal confidence sets. We plot the results for the logit scores, distance transformed scores (DT) and boundary box scores (BB) from left to right. The bounding box scores are discontinuous which is the cause of the discreteness of the rightmost histograms.

### A.6.9 COMPARING THE PROPORTION

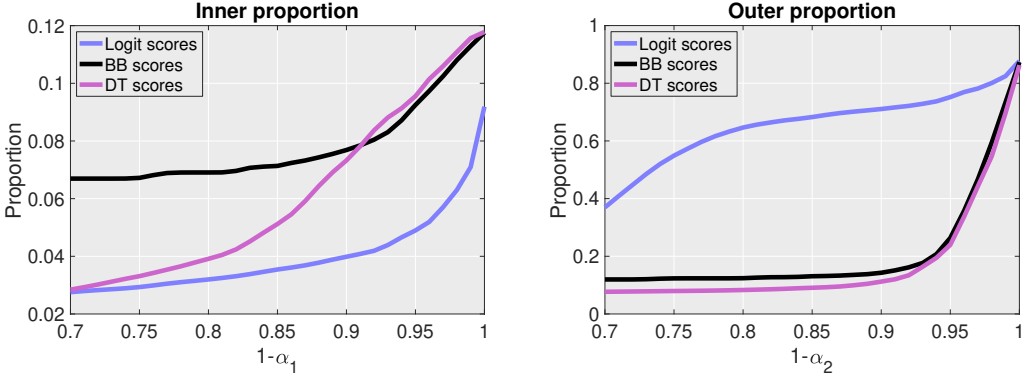

Figure A17: Measuring the proportion of the entire image which is under/over covered by the respective confidence sets. Left: proportion of the image which lies within the true mask but outside of the inner set. Right: proportion of the image which lies within the confidence set but outside of the true mask. For both a lower proportion corresponds to increased precision.

## A.7 Additional settings for brain mask segmentation

### A.7.1 Comparing original and distance transformed scores on the learning dataset

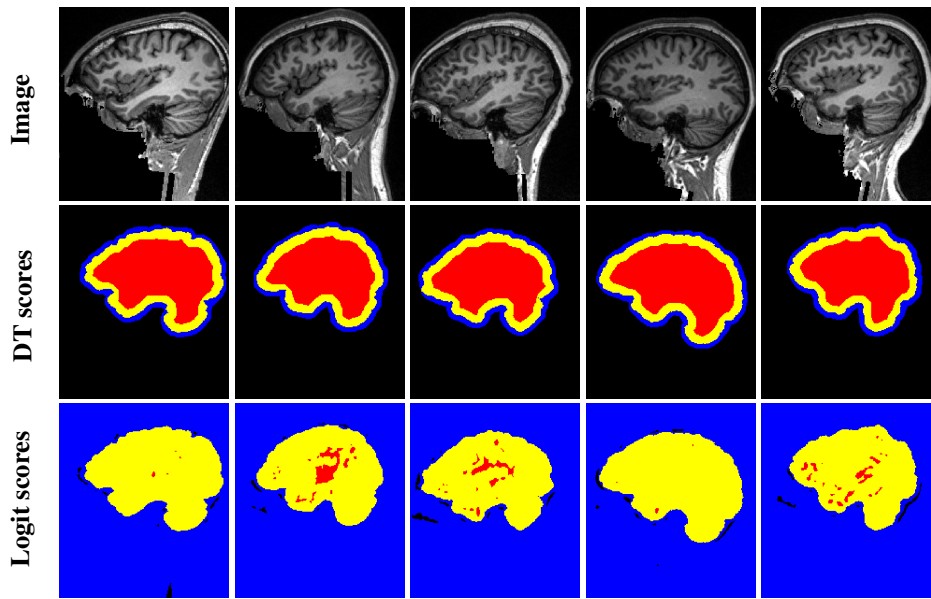

Figure A18: *Learning the best transformation for brain mask segmentation. First row: original images from different subjects. Second row: confidence sets provided by calibrating the distance transformed scores on the learning dataset. Third row: confidence sets produced using the logit scores on the learning dataset. Using the logit scores produced uninformative confidence sets. Instead the distance transformation is a big improvement.*

A.7.2 COMPARING SMOOTH TRANSFORMED SCORES ON THE LEARNING DATASET

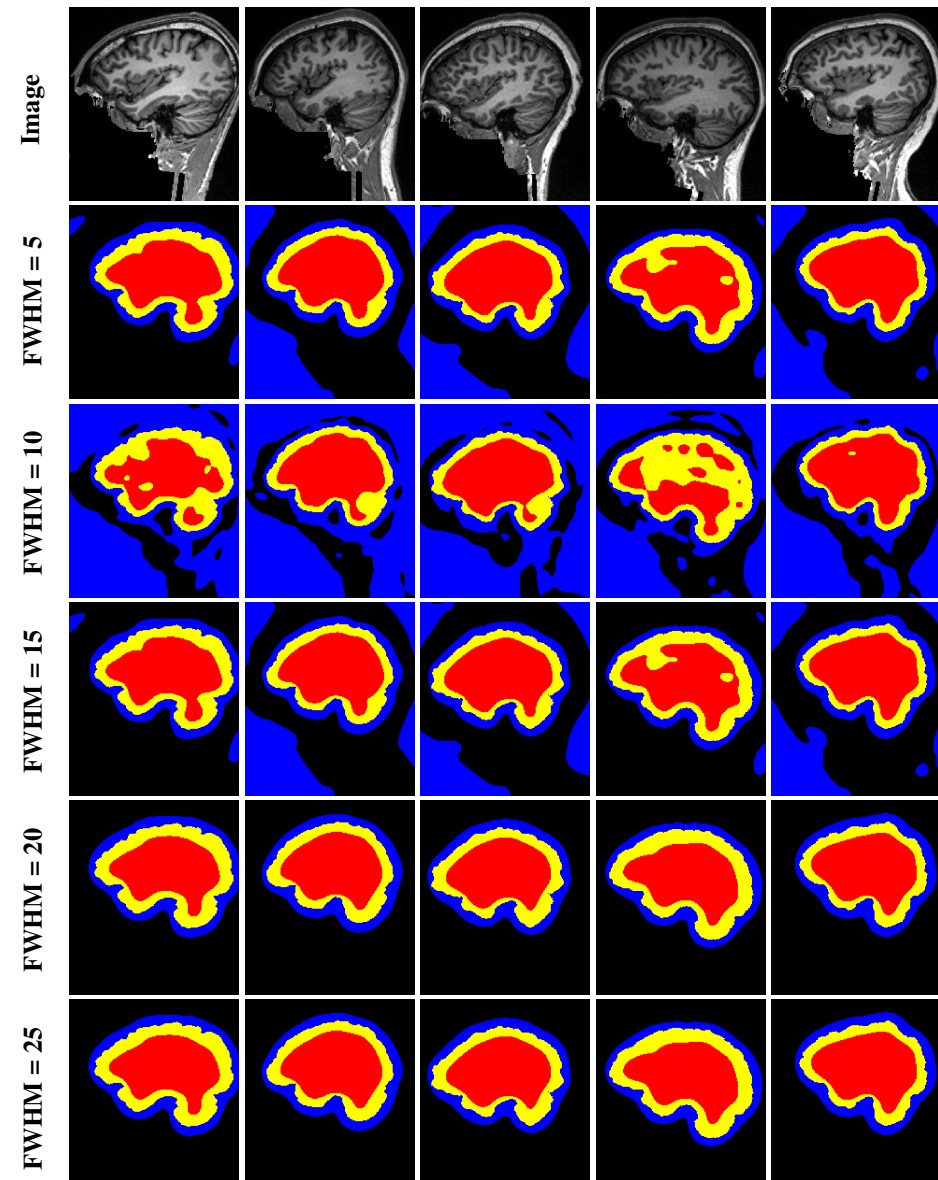

Figure A19: *Inner and outer sets computed by comparing smooth score transformations on the learning dataset. Scores were smoothed using an isotropic Gaussian kernel with full width at half maximum (FWHM) taking values in $\{5, 10, 15, 20, 25\}$ mm. The resulting inner and outer sets based on increasing levels of applied smoothness are shown from top to bottom. The performance appears to peak at 20mm.*

### A.7.3 COMPARING ORIGINAL AND DISTANCE TRANSFORMED SCORES ON THE TEST DATASET

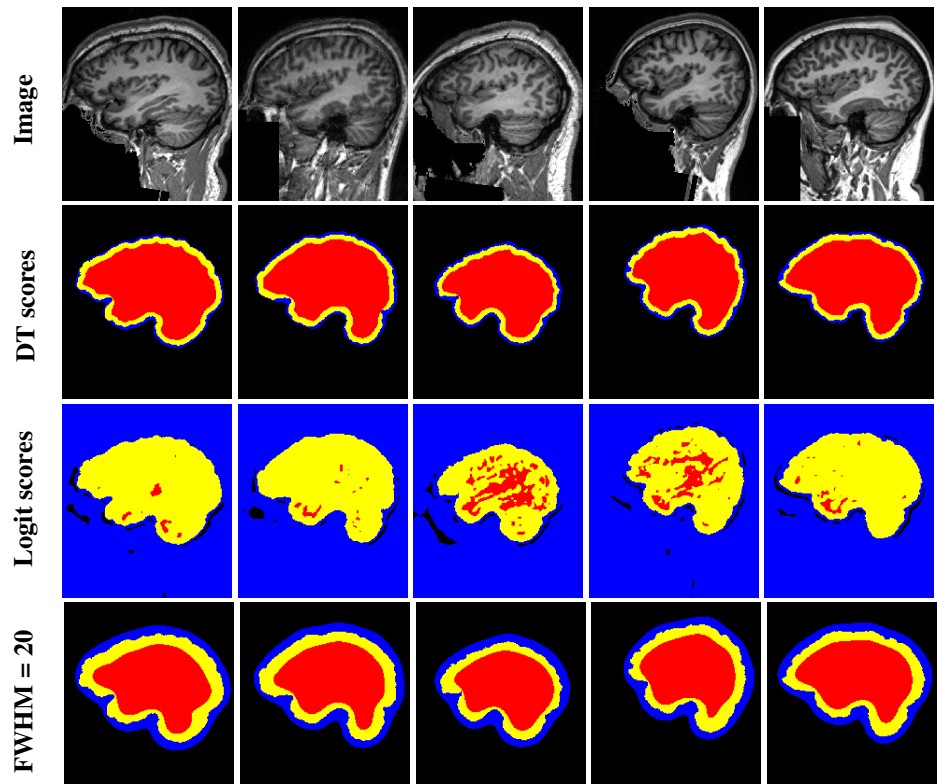

Figure A20: *Inner and outer confidence sets for brain mask segmentation using the distance transformed and logit scores. Top row: brain images for each subject. 2nd row: the inner and outer confidence sets produced by calibrating the distance transformed scores on the calibration dataset. 3rd row: the inner and outer confidence sets produced by calibrating the logit scores on the calibration dataset. 4th row: the inner and outer confidence sets produced by calibrating the smoothed scores (smoothed with an isotropic Guassian kernel of 20mm - chosen for comparison here because it performed the best on the learning dataset out of the smoothing levels considered). As for the learning dataset the logit scores perform very poorly and are not able to separate the background from the segmented masks with confidence. Instead the distance transformed scores do a very good job at segmenting the mask. Indeed they do slightly better on the calibrated dataset than on the learning dataset. This occurs as the learning dataset is relatively small and does not capture the full picture. The smooth scores improve on the logit scores but do not provide as tight bounds as the distance transformed scores for neither inner nor outer sets.*

### A.7.4 COMPUTING THE COVERAGE FOR THE BRAIN IMAGING DATA

*In order to study the coverage rate of the methods in the context of the brain imaging application we peform a similar validation to that described in Section 3.3 for polyps segmentation. To do so we divide the 474 subjects left, after excluding the learning dataset, into 300 calibration and 174 test images. We do this 1000 times, randomly sampling the sets of 300 and 174 images respectively and measuring the coverage in each run. We average the coverage over the 1000 runs and display the results in Figure A21. Note that unlike the box scores considered in Section 3.3, the smooth scores are not discrete so do not display discreteness issues at lower levels of coverage.*

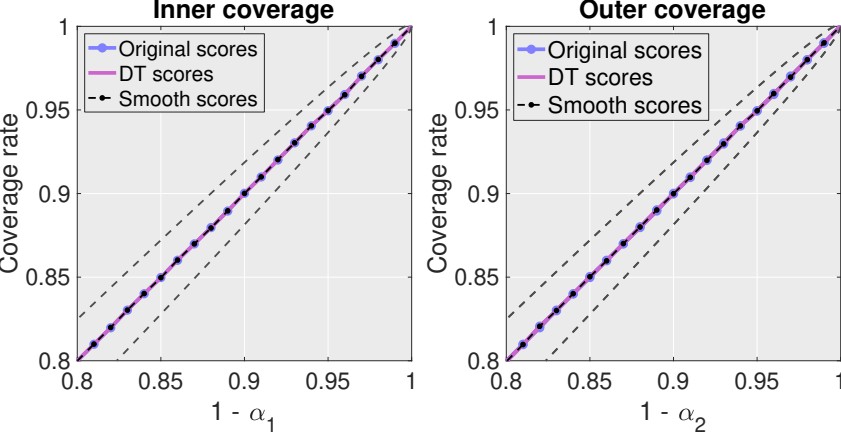

Figure A21: *Coverage levels of the inner and outer sets averaged over 1000 validations for the original, distance transformed (DT) and smoothed scores (smoothed with a full width at half maximum of 20mm). The nominal rate is acheived in all settings considered.*

## A.8 ADDITIONAL SETTINGS FOR TEETH SEGMENTATION

### A.8.1 COMPARING ORIGINAL AND DISTANCE TRANSFORMED SCORES ON THE LEARNING DATASET

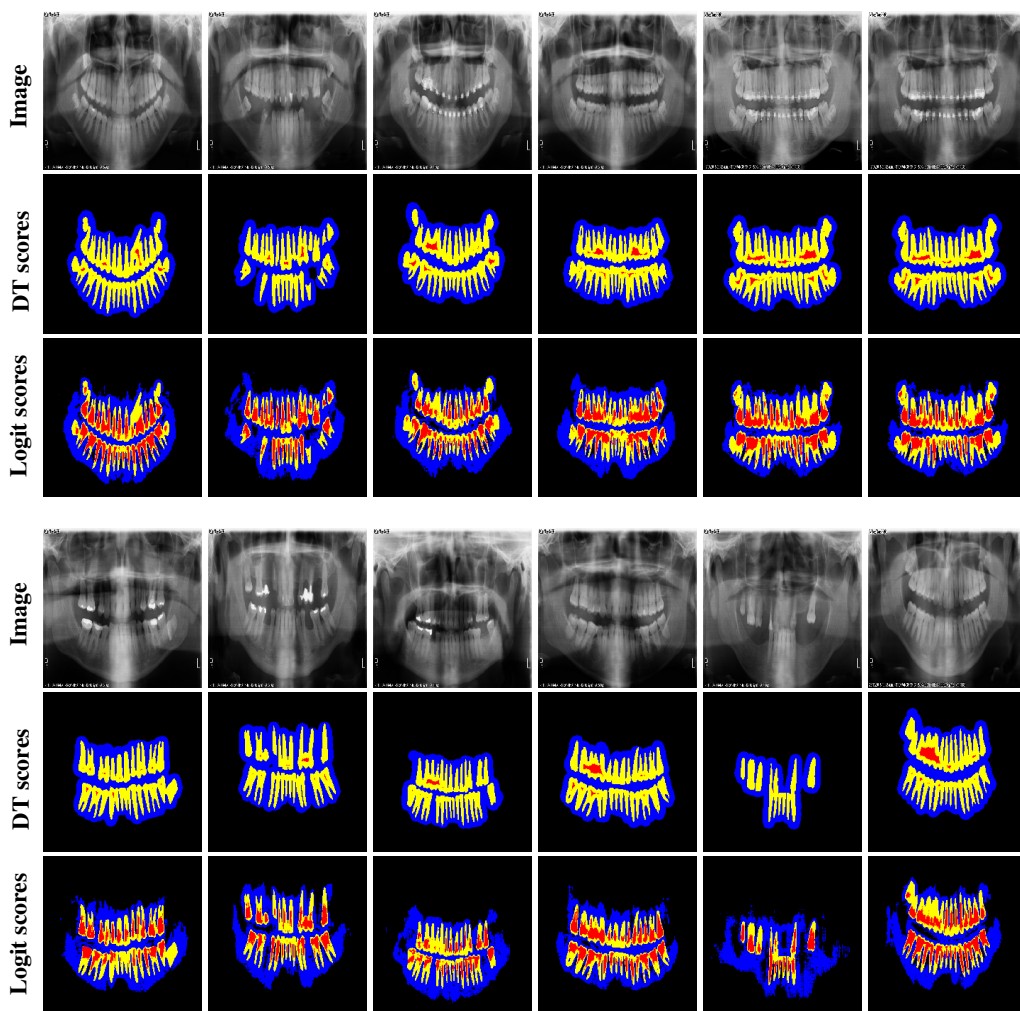

Figure A22: *Inner and outer confidence sets for brain mask segmentation calibrated and plotted on the learning dataset. 12 images are plotted in two sets of 6, for each set the rows are as follows. First row: original images. Second row: results of distance transformed scores - providing tight outer sets but uninformative inner sets. Third row: logit scores providing looser outer sets but more informative inner sets though these can be improved by smoothing see Figure A23.*

A.8.2 COMPARING SMOOTH TRANSFORMED SCORES ON THE LEARNING DATASET

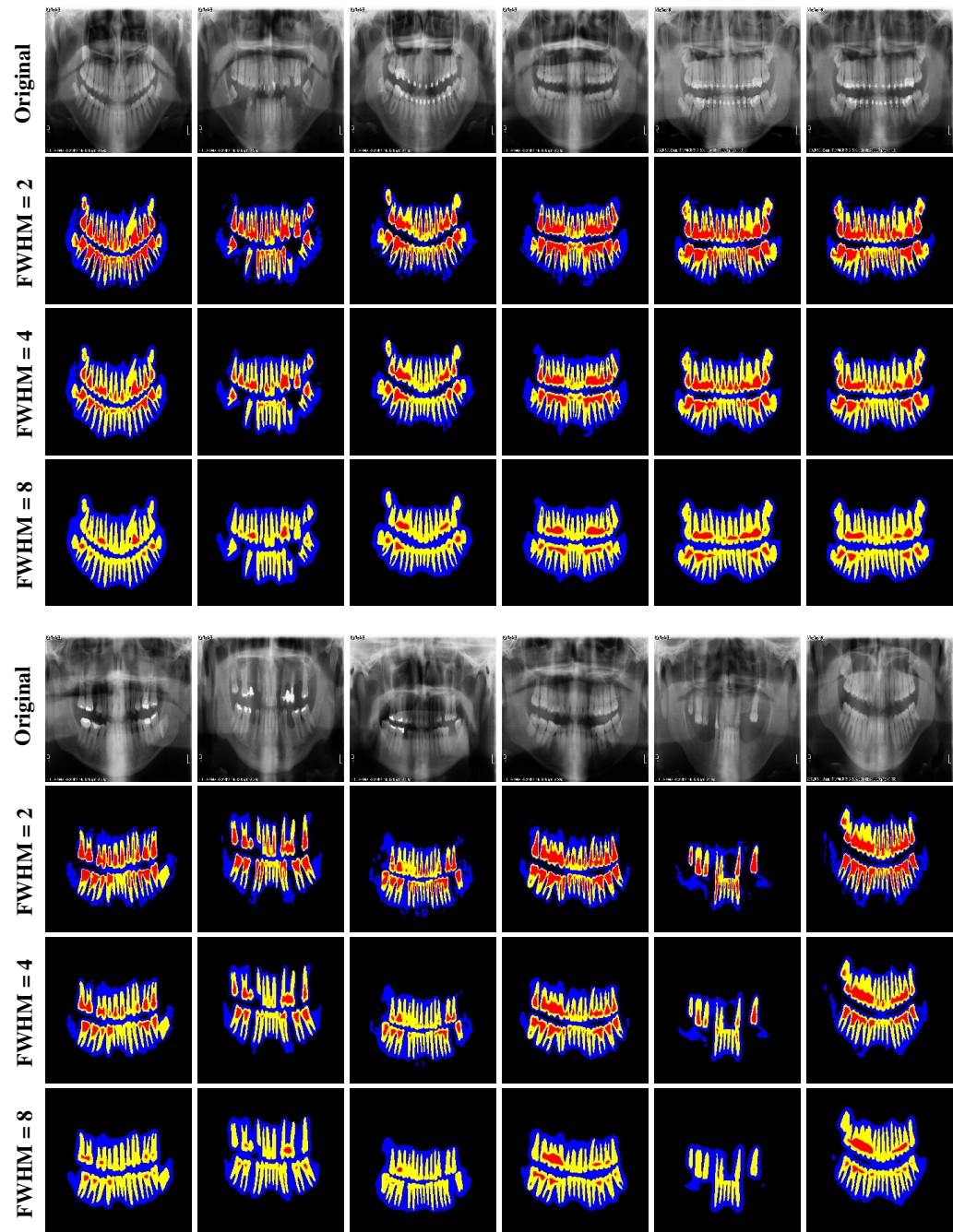

Figure A23: *Inner and outer sets computed by comparing smooth score transformations on the learning dataset. Scores were smoothed using an isotropic Gaussian kernel with full width at half maximum (FWHM) taking values in $\{2, 4, 8\}$ pixels. For each set of 6 images the resulting inner and outer sets based on increasing levels of applied smoothness are shown from top to bottom. A FWHM of 2 pixels is the best for the inner set and indeed performs better than the logit scores shown in Figure A22. Instead an increased level of smoothness is better for the outer set, attaining comparable performance to the distance transformed scores though with some additional blobs.*

### A.8.3 COMPARING TO THE LOGIT SCORES ON THE TEST DATASET

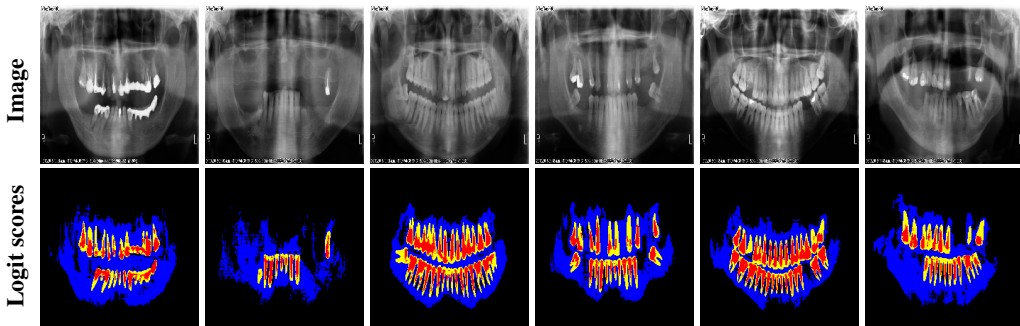

Figure A24: *Inner and outer confidence sets for brain mask segmentation performed on the test set computed by calibrating and predicting using the logit scores. The performance is less good than the score transformations optimized on the learning dataset and shown in the main text and has been included here for reference. The outer sets are larger and less precise than those based on the distance transformation. The inner sets are good but not quite as good as the ones based on a small amount of smoothing.*

### A.8.4 COMPUTING THE COVERAGE FOR THE TEETH DATASET

*In order to study the coverage rate of the methods in the context of the brain imaging application we peform a similar validation to that described in Sections 3.3 and A.7.4. To do so we divide the 198 subjects, into two subsets of size 99. We do this 1000 times, randomly sampling the subsets of 99 images respectively, calibrating on the first subset and measuring the coverage on the second in each run. We average the coverage over the 1000 runs and display the results in Figure A25.*

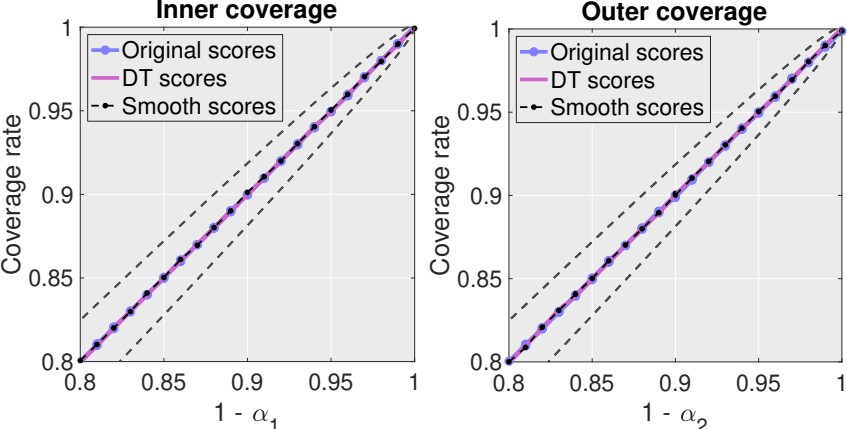

Figure A25: *Coverage levels of the inner and outer sets averaged over 1000 validations for the original, distance transformed (DT) and smoothed scores (smoothed with a full width at half maximum of 2 pixels). The nominal rate is acheived in all settings considered.*

## A.9   COMPARING PERFORMANCE METRICS FOR EACH SEGMENTATION MODEL

*In the table we display performance metrics for each model, computed over the validation set used in each data application.*

Table 1: *Performance Metrics over the validation set*

| Model | Application | Average Dice Score | Average Precision | Average Recall |
|---|---|---|---|---|
| PraNet | Polyps | 0.869 | 0.877 | 0.881 |
| HDBET | Brain imaging | 0.976 | 0.961 | 0.992 |
| U-Net based GAN | Teeth | 0.933 | 0.935 | 0.932 |

## A.10   RELATIONSHIP WITH MULTIPLE TESTING ERROR RATES

### FAMILY-WISE ERROR RATE (FWER)

*In traditional multiple hypothesis testing Family-Wise Error Rate (FWER) is the probability of making at least one false discovery across a set of considered hypotheses. Given a multiple testing problem in which $m$ hypotheses are tested and a multiple testing algorithm $\mathcal{M}$, let $V(\mathcal{M})$ denotes the resulting set of false discoveries, $R(\mathcal{M})$ the set of rejected hypotheses and $T$ be the set of true rejections. Then the FWER is defined as:*

$$FWER(\mathcal{M}) := \mathbb{P}(|V(\mathcal{M})| \geq 1).$$

*Then, given $\alpha > 0$, if we can guarantee that FWER $\leq \alpha$ then it follows that $R(\mathcal{M}) \subseteq T$ with probability at least $1 - \alpha$. This statement is thus analogous to the coverage guarantees which we provide in Theorems 2.1 and 2.2 in the sense that a probabilistic gaurantee on the inclusion probability is provided.*

### FALSE DISCOVERY RATES AND PROPORTIONS

*Instead the false discovery proportion in the multiple testing setting for an algorithm $\mathcal{M}$ is given by:*

$$FDP(\mathcal{M}) := \frac{|V(\mathcal{M})|}{|R(\mathcal{M})|} \cdot \mathbf{1}_{|R(\mathcal{M})|>0}$$

*and the False Discovery Rate is given by:*

$$FDR(\mathcal{M}) = \mathbb{E}\left[FDP\right],$$

*where $\mathbf{1}_{|R(\mathcal{M})|>0}$ is an indicator function that is 1 when $|R(\mathcal{M})| > 0$ and 0 otherwise. Controlling the FDP in probability is thus analogous to the proportion of the true mask that lies within the discovered sets, as in Bates et al. (2021) whilst controlling the FDR is instead analogous to the risk control discussed in Angelopoulos et al. (2021) in which conformal inference is used to control the expected proportion of the true mask which is discovered.*

