# OpenReview forum: "Conformal confidence sets for biomedical image segmentation"
_ICLR.cc/2025/Conference — Submitted to ICLR 2025_

### Official Review · Reviewer_XxS9 · 2024-10-28

**Soundness:** 4
**Presentation:** 3
**Contribution:** 3
**Rating:** 8
**Confidence:** 4

**Summary:**

The authors propose a conformal prediction method that computes confidence sets with spatial uncertainty guarantees in image segmentation from any machine learning model. They illustrate the usefulness of the proposed method on medical images.

**Strengths:**

The paper is well-written and clear, although it took a second read-through to fully understand. The proposed method seems to work very well, and the presented experiments are convincing.

**Weaknesses:**

I am missing more quantitative results. For instance, aggregated coverage scores (e.g., mean; or other metrics, e.g., evaluate Equations 1 and 2) for the different versions on more than one dataset. This comparison should then also include some existing methods, to illustrate the relative strengths of different methods.

As just mentioned, for the results to be more convincing, I would also like to see examples on more than just one dataset.

Also, there must be other score transformation functions that could also be evaluated. Testing a couple more could strengthen the results and make it more convincing.

**Questions:**

- Couldn't a related/similar smooth distance be defined using kernels?
- What is called "original scores", is this when you use the identity score transformation?
- What are the dashed lines in Figures 4 and 5?

Major comments:
- Add labels and/or legends to the rows and columns of the figures.

Minor comments:
- The word "polyp" is misspelled in different ways in almost every instance. Do check this.
- It says "... the set a side [num] images ...", or something similar, a few times. Check the grammar there.

---

> ### Author Response · Authors · 2024-11-20
>
> We are very pleased that the reviewer enjoyed reading the paper and are grateful for their comments and questions which we address below.
>
> We have taken the reviewer’s advice on board and, in order to improve the quality of the manuscript, have included applications to two new datasets involving segmentation in the context of brain imaging and dentistry. Our results on these datasets show the robustness and wide applicability of our approach. See the relevant Section 4, 5, A.7 and A.8 of the updated paper for the results and application examples.
>
> Regarding the need for quantitative metrics we have now included dice, precision and recall metrics, in Section A.9, for the 3 different segmentation models used in the paper. These metrics correlate with the performance of the distance transformed scores but not necessarily with other score transformations. Moreover we would like to clarify that evaluation of the inclusion specified in equations 1 and 2 is done in the validations in Section 3.3 (and for the new datasets in Sections A.7.4 and A.8.4). These validations subsample the data with replacement (each time dividing into a calibration and a test set) and check whether the inclusions 1 and 2 hold in order to establish that the methods have the right coverage rate. They show that for each of the datasets considered the confidence sets provide coverage at the nominal rate for interesting coverage levels.
>
> In the first version of the paper we compared to the bounding box approach of [1] as this is the main other approach we are aware of which controls the same error rate. Other methods used in conformal image segmentation typically consider weaker error rates as these are easier to satisfy whilst being less meaningful. However score transformations such as the distance transformation can be very helpful when using these other methods for the same reasons they are helpful in our context. We shall prepare and include an illustration of the resulting benefits of doing so, for other methods such as conformal risk control [2], for the final version of the paper. We shall also show quantitatively the degree to which sets derived using risk control (with the expected proportion of false non-discoveries) provide (severe) undercoverage when considering inclusion coverage rates.
>
> [1] Andéol, Léo, et al. "Confident Object Detection via Conformal Prediction and Conformal Risk Control: an Application to Railway Signaling." Conformal and Probabilistic Prediction with Applications. PMLR, 2023.
>
> [2] Angelopoulos, Anastasios N., et al. "Conformal risk control." ICLR, 2024.
>
> We agree that there are other score transformations which can be considered. In particular as the reviewer remarks smoothing the score contributions via a smoothing kernel is a good idea. We illustrate this in the new applications to brain imaging and dental records, see Sections 4, 5, A.7 and A.8. Here we compare the results of smoothing the scores using a Gaussian kernel with varying levels of applied smoothness. In the brain imaging application we see that this leads to a big improvement over the use of the original scores (which perform quite poorly). However in this setting the improvement is not as great as using the distance transformed scores. Instead for the dental application, smoothing is very helpful and in fact provides the largest inner confidence sets, which we then use in practice. For this application it also helps to provide tight outer sets. These can in fact be tighter than those provided by the distance transformation however tend to have extra blobs which do not correspond to teeth which is why we settled on the distance transformation for the final calibration.
>
> Instead for the polyps application we found that smoothing did not significantly improve the quality of the inner and outer sets on the learning dataset, likely because the score contributions from the model are already smooth (see e.g. the surface plot of the scores in Figure 2). We will add the results of applying smoothing in the polyps application to the final version of the manuscript.
>
> We have added labels to the rows/columns of the figures displaying the confidence sets throughout the main text and the appendix and thank the reviewer for this suggestion as it greatly helps to improve the clarity. Moreover we would like to apologize for the spelling error of polyps which we have now corrected in the updated draft and appreciate that this was spotted. We have also replaced "... the set a side [num] images ...", with the “… [num] images which we set aside” or another appropriate variant.

---

> ### Author Response · Authors · 2024-11-20
>
> Regarding the reviewers remaining questions. What we referred to as the original scores are indeed the scores which result from using the identity transformation. In order to improve the clarity of this in the paper we now refer to these scores as the logit scores or the untransformed logit scores throughout the paper instead of as the original scores. Furthermore the dashed lines in Figure 4 provide 95% uncertainty bands for the coverage, we have now clarified this in the caption of Figure 4. Instead the grey dashed line in Figure 5 indicates the value 1 at all levels, this is included for comparison because the best possible value of the inner and outer ratio in the respective plots is 1.
>
> We thank the reviewer once more for their helpful comments and look forward to hearing their thoughts on our response and discussing any follow up questions that they may have.

---

> > ### Comment · Reviewer_XxS9 · 2024-11-22
> > **Follow-up comments and clarifications**
> >
> > Thank you for carefully addressing my comments and concerns. I still have the following comments:
> >
> > - I didn't mean metrics on the segmentation performance, but metrics on the uncertainty quantification. It would be great with aggregated metrics saying something about the overall performance for each method, to summarise their relative strengths and weaknesses. Or is this what you mean by the inclusion of risk control of the coverage?
> > - About score transformation: It is relevant to smooth the score contributions, but what I was thinking of was a variant of the score transformation that used a kernel instead of the sign function to measure similarity between v and a set A. Like a soft version of the current score transformation.
> > - Theorem 2.8: The H should have a subscript \rho, with the distance metric.
> > - Line 269: Says "poplys".
> > - Figure 2: Label the rows of the image grid.
> > - Line 364: Missing start of the sentence, or just that the first letter should be capital?
> > - Figure 5: It still says "Original scores".

---

> > > ### Author Response · Authors · 2024-11-23
> > >
> > > We would like to thank the reviewer for their further comments and for the effort they have put into reading and helping to improve the paper.
> > >
> > > Regarding the quantitative metrics on uncertainty quantification. If the reviewer doesn’t mind would they be able to clarify more specifically which metrics in particular they would like to see. We interpreted the mean aggregated coverage score, discussed in the reviewer’s first comments to be the same as the evaluation of equations (1) and (2) mentioned by the reviewer. As we clarified above these evaluations are included in the text in Figure 4 (for the polyps applications) and in Figures A21 and A25 for the brain imaging and teeth segmentation examples. Is the reviewer referring to alternative uncertainty metrics or are these sufficient? We would be very happy to include other measures which quantify the uncertainty. We included in Figures 5 and A17 measures of the performance among the different score transformations however we could also consider other measures to compare such as the width of the confidence bands or alternative measures.
> > >
> > > Regarding the comparison to other methods. This is indeed what we meant by the inclusion of risk control, i.e. we shall calculate equations (1) and (2) for sets designed for risk control [1] and other existing conformal inference methods – these will under cover because they are not designed to control the coverage but instead an alternative error rate. This will help to explain the strengths of using confidence sets over these alternative approaches.
> > >
> > > [1] Angelopoulos, Anastasios N., et al. "Conformal risk control." ICLR, 2024.
> > >
> > > Thank-you for clarifying the type of transformation which you meant. It would also be interesting to explore this. We will add an exploration of the impact of using different kernel functions, used to measure similarity between v and a set A, and test how these perform relative to our existing transformations for the final version of the paper.
> > >
> > > We will also label each of the rows of Figure 2 with the labels “Scores” and “Confidence Sets” on the right hand side of the figure (we haven’t yet managed to do so due to a small latex issue but will do so when we figure that out).
> > >
> > > With regards to the other points of the reviewer we have now resolved these and we thank the reviewer for pointing these out.

---

> > > > ### Comment · Reviewer_XxS9 · 2024-11-25
> > > >
> > > > Thank you for clarifying about Figure 4 and Figure A21 and A25, I had misunderstood what was in those. Yes, that was what I was looking for, so all good.
> > > >
> > > > As you mentioned, about confidence band width, it would probably be interesting to measure the Hausdorff distance (for instance) between the inner and outer sets, but that would not be required from my side.

---

> > > > > ### Author Response · Authors · 2024-12-03
> > > > >
> > > > > We thank the reviewer for their thoughts. We shall aim to include a measure of the Hausdorff distance between the inner and outer sets for the final version of the paper.

---

### Official Review · Reviewer_8vPV · 2024-10-29

**Soundness:** 2
**Presentation:** 2
**Contribution:** 1
**Rating:** 3
**Confidence:** 4

**Summary:**

The paper proposes a conformal prediction based method to quantify the uncertainty for medical image segmentation. The proposed method is particularly designed for pre-trained segmentation models which notoriously make overconfident and wrong predictions. The proposed method learns thresholds using the maximum logit scores from a calibration set for the inside and outside of the ground truth masks and apply them on the logit scores of the test image to return conformalized segmentation prediction which guarantees to include the ground truth segmentation. The paper shows that naively learning the outside thresholds on max logits is not optimal and propose to transform the scores using a distance to make sure that far away pixels have lower scores. The method is validated on a single dataset for polyp segmentation and the results show that the proposed method produces conformal sets with narrower boundaries compared to using scores which are not transformed.

**Strengths:**

- The idea of using transformed max logit scores is simple but quite effective strategy to produces conformal segmentation sets.
- The presented experiments show the effectiveness of the method compared to using non-transformed logits.

**Weaknesses:**

1- Although I found the proposed idea of transforming max logit scores interesting, I don't think that the paper presents enough contribution to be presented in ICLR. The idea of applying conformal prediction to max logits for inside and outside of the boundaries is a direct extension of initial conformal prediction methods developed for segmentation, and applying transformations based on distance is an intuitive choice to refine predicted boundaries.

2- The paper does not present any comparisons with the existing conformal prediction works for image segmentation.

[1] Mossina et al. Conformal Semantic Image Segmentation: Post-hoc Quantification of Predictive Uncertainty, CVPR Workshops, 2024,

3- The method is evaluated on only a single dataset. Multiple datasets should be included to make sure that the performance generalizes across datasets.

4- In many segmentation tasks, we are interested in segmenting multiple structures. The paper only focuses on binary segmentation. I think the method should be validated on multi-class setting to make sure that it is also applicable in that setting.

5- The explanation of how the method is applied at test time could also be clearer. As I understand it, during testing, the method applies the inner threshold on max logits to find inner boundaries, then applies a distance transformation based on each pixel’s distance from these inner boundaries, and finally applies an outer boundary threshold. However, the exact steps of the algorithm during test time need more clarification.

6- In conventional uncertainty quantification algorithms for segmentation such as [2, 3] the uncertainty is quantified by the variance of the segmentation samples generated from the posterior distribution. How can the quantification be done in this case? Is it the margin between the inner and outer boundaries? Is the uncertainty quantified by the algorithm correlates with the uncertainty in the input image? For example, does the method output larger margins when there is greater disagreement between the segmentations of different experts?

[2] Kohl et al. A Probabilistic U-Net for Segmentation of Ambiguous Images
[3] Erdil et al. MCMC Shape Sampling for Image Segmentation with Nonparametric Shape Priors

7- The margin between the inner and outer boundaries appears quite large and there can be many unplausible segmentations within this area. For practical applications, an uncertainty quantification method should ideally produce a set of plausible segmentation samples within this margin, rather than simply indicating a large margin that may or may not include the ground truth segmentation. How could one obtain a plausible segmentation sample from this margin?

**Questions:**

- How does the results generalize to other datasets and segmentation of multiple structures?
- How does the uncertainty quantified by the proposed method relates with the real uncertainty (assuming it can be measured by the disagreement between multiple experts)?
- How one can use the proposed method in a practical application? Can we get samples of plausible segmentations within the margin outputted by the algorithm?

---

> ### Author Response · Authors · 2024-11-20
>
> We are grateful for the comments and thoughts of the reviewer and for the opportunity to clarify our contributions.
>
> 1- The distance transformation is indeed a sensible choice of score transformation. However as far as we are aware other papers have not considered it in the context of conformal inference for image segmentation. Given how necessary this transformation turns out to be in some applications (see e.g. the new brain imaging example in which the untransformed scores provide extremely uninformative bounds), to us this is an important gap to fill in the literature. We also regard the theory which we derive surrounding inner and outer sets, including the newly added results, Theorems 2.8 and A.4, as a key contribution.
>
> 2 - We would like to clarify further that we in fact do compare to the results of other existing methods. In particular the bounding box approach of [1] is compared to on the learning dataset and the testing datasets, for the polyps application, and shown to perform less well than the use of the distance transformation. This is shown visually in Figure 2 and Figures A8-12. We also compared to the precision of this approach in Figure 5 and included it in our validations in Figure 4. We explain the relationship with [1] in Section 2.5 of the manuscript.
>
> [1] Andéol, Léo, et al. "Confident Object Detection via Conformal Prediction and Conformal Risk Control: an Application to Railway Signaling." Conformal and Probabilistic Prediction with Applications. PMLR, 2023.
>
> Our existing results in fact also compare to the result of applying [2], the paper mentioned by the reviewer. This is because for our problem setting the approach of [2] is equivalent to empirical risk control [3] with the binary loss function which we showed can be used to derive valid inner and outer sets in Section A.2. We have clarified this in Remark 2.4. Applying the method of [2], without modification, in our context would result in the blue outer set obtained from the identity score transformation which is typically very wide and not useful. This is exemplified in the brain imaging application, see Figure A20, in which the blue outer set (which would be the result of applying the algorithm in [2]) obtained from using the untransformed scores is extremely uninformative. Indeed [2] observed very poor performance with the binary loss function, noting that the resulting “prediction set will be theoretically valid but not very informative”.  The use of the score transformations and the distance transformation in particular is thus crucial in improving the width of the confidence sets. As far as we are aware our paper is the first (other than the bounding box approach of [1] which we compare to) to provide informative conformal confidence sets which are guaranteed to fully contain the segmented outcome (rather than controlling another weaker error rate),
>
> [2] Mossina et al. Conformal Semantic Image Segmentation: Post-hoc Quantification of Predictive Uncertainty, CVPR Workshops, 2024,
>
> [3] Angelopoulos, Anastasios N., et al. "Conformal risk control." ICLR, 2024.
>
> 3 - We have now included two additional applications involving brain imaging and dentistry. These show that the performance of the model indeed generalizes across datasets. It also helps to emphasize the need for score transformations. The distance transformation does particularly well on the brain imaging dataset. We have performed validations on these datasets, see Sections A.7.4 and A.8.4, which show that the model correctly controls the coverage rate in these settings.
>
> 4- Regarding the reviewer’s question about segmentation of multiple structures. This is indeed an interesting question. The segmentation problem for each one of these multiple structures is itself a binary segmentation problem. As such corresponding results for multiple structures follow as a corollary to our results. Joint coverage over the structures can then be obtained by jointly sampling the maximum of the scores over the different classes. We shall formalize this and add an application for the final version of the paper.
>
> 5- In order to clarify what the algorithm does during test time we have included a formal algorithm describing the steps taken by the model. See Algorithm 1 in Appendix A.5, now referenced in Section 3.2. Inner and outer thresholds are in fact computed separately based on the inner and outer scores respectively during calibration. When applying the distance transformation the distance is computed relative to the predicted mask obtained by thresholding the logit scores at 0 not to the inner set. Then at test time transformed inner and outer scores are calculated and compared to the calculated threshold. We hope that the provided algorithm helps to make the steps taken clearer.

---

> ### Author Response · Authors · 2024-11-20
>
> 6- Uncertainty quantification for our method is indeed quantified by the margin between the inner an outer confidence sets. We do not rely on a posterior distribution, instead using the calibration set to calculate the inner and outer thresholds. As such our method does not make assumptions on the distribution of the data in order to provide valid uncertainty.
>
> In particular the width of the confidence bands directly depends on the quality of the neural network. I.e. as the predicted segmented mask approaches the ground truth mask in Hausdorff distance both inner and outer sets will converge to the ground truth mask. In order to formalize this we have added Theorems 2.8 and A.4 which show that if the Hausdorff distance between predicted and ground truth masks on the calibration dataset is bounded then confidence sets for new observations are precise. Importantly this result does not hold for the original untransformed scores which can give very wide and uninformative confidence sets even when the neural network provides very good predictions. This very well illustrated in the brain imaging application, see Figure A20 in Appendix A.7.
>
> In the case that experts disagree on the true segmented mask we would for now recommend using a consensus mask which is a function of the masks produced by each expert. In that case the method would provide confidence bands relative to this consensus mask. The method is only as good as the quality of the expert calculated masks and relies strongly on a good quality ground truth. We do not directly incorporate the uncertainty in the ground truth masks in our approach but it would be very interesting to do so, as we now observe in the discussion.
>
> 7- The size margin between the inner and outer boundaries (for the confidence sets obtained from using the distance transformed scores) depends on the application setting and quality of the image segmentation algorithm, as shown in Theorems 2.8 and A.4 and discussed in the response to (6) above. The width of the uncertainty bands helps to visually capture the uncertainty of the model and in our view allows practitioners to better understand the limitations of these models.
>
> It is indeed the case that not all segmentations within the margin will be equally plausible. Because we are not working with a posterior distribution it is not possible to obtain samples from the model. Instead obtaining a set of plausible segmentations within these bounds would in our view require additional biological information to be taken advantage of. We have added a comment to the discussion on this point as an interesting direction for future research.
>
> ------------------------------------------------------------------------------------------------------------
>
> We would direct the reviewer to our responses to 3 and 4 (and the response to all reviewers) with regards to their first question, to 6 for the second question and thirdly to 5 and 7 and below for the response to their third question.
>
> Regarding how the method can/should be used in practice. This depends on the application setting. For polyps segmentation the method could be used to rule over regions of the image where the polyps could lie. We can be sure, up to the guarantee provided by the model that there are no polyps outside of the blue set meaning that practitioners could deprioritize looking for polyps within those regions.
>
> Instead for instance in the brain imaging application it is important to detect locations which lie within/outside the brain for follow up analyses. Within the inner set we can be sure to find areas inside the brain which could help with alignment further down the pipeline and the detection of activation (e.g. when using fMRI). Instead the outer set can be used to mask out areas where we can be sure that there is no brain, and thus no activation. Having precise confidence bounds on this is important because otherwise we risk missing areas of the brain.
>
> We thank the reviewer once more for their helpful comments and look forward to hearing their thoughts on our response and discussing any follow up questions that they may have.

---

> > ### Comment · Reviewer_8vPV · 2024-11-22
> >
> > Thank you to the authors for their detailed response. I truly appreciate the effort they put into providing explanations and conducting additional experiments. However, I still have some concerns that prevent me from improving my initial score.
> > 	1.	My concern regarding the contribution of the paper remains. While I understand that distance transformations have not been explored before in the literature, I still view this as a relatively minor contribution. I would be more convinced if the practical value of the method was demonstrated more clearly. Currently, I struggle to see how such a method can be effectively applied in practice, which leads me to my second point.
> > 	2.	The authors propose the following practical use case for their method:
> >
> > 	“For polyp segmentation, the method could be used to rule over regions of the image where the polyps could lie. We can be sure, up to the guarantee provided by the model, that there are no polyps outside of the blue set, meaning that practitioners could deprioritize looking for polyps within those regions.”
> >
> > While I understand this scenario, I am not convinced that it significantly reduces the time and effort required by practitioners or addresses a pressing problem for them. In all the images shown in the experiments, there is a single polyp, and the images are relatively small. Practitioners can already identify the polyp region without needing to extensively search the entire image. I would find the method more compelling if there were examples involving larger images where experts or practitioners genuinely face challenges in locating polyps across a wide area. In such cases, narrowing the search region could provide substantial benefits. The same critique applies to the teeth dataset. Experts (and even non-experts) already know where to look to find the teeth. If the conformal prediction approach merely provides smaller regions to search within—leaving practitioners to “manually” segment those areas—it is unclear to me what significant advantage this offers in a real-world scenario.
> >
> > For these reasons, I am inclined to maintain my initial rating.

---

> ### Author Response · Authors · 2024-11-23
>
> We would like to thank the reviewer once again for their feedback. We admit that we had been hoping that the reviewer would consider increasing their score slightly in light of the changes which we have made. The addition of the new applications and theory – prompted by the helpful comments of the reviewer (and other reviewers) – have in our view greatly strengthened the paper.
>
> With regards to the reviewer continued concerns. We shall address these in reverse order.
>
> With regards to the reviewer’s second point. It is our strongly held view that deep learning models are widely used without proper uncertainty quantification. Indeed as discussed in [4], “neural networks do not deliver certainty estimates or suffer from over- or under-confidence”. These models are powerful but are essentially black boxes the outputs of which are difficult to properly understand. We would strongly argue that whenever it is useful to use a deep learning based segmentation model it is important to have proper confidence bands on the output in order to better understand the limitations of the model. As discussed in [5], “…medical AI, especially in its modern data-rich deep learning guise, needs to develop a principled and formal uncertainty quantification (UQ)”. Deep learning models are widely used without much thought or consideration of uncertainty and in our view the field is in great need of a change. We regard our work as a step along this path in the right direction.
>
> [4] Gawlikowski, Jakob, et al. "A survey of uncertainty in deep neural networks." Artificial Intelligence Review 56.Suppl 1 (2023): 1513-1589.
>
> [5] Begoli, Edmon, Tanmoy Bhattacharya, and Dimitri Kusnezov. "The need for uncertainty quantification in machine-assisted medical decision making." Nature Machine Intelligence 1.1 (2019): 20-23.
>
> The reason for which uncertainty quantification is necessary in deep learning is the same as the reason for which statisticians advise against providing point estimates on effect sizes without confidence bands. In our setting the predicted output of the neural network is the point estimate of the segmented outcome and our confidence bands provide the necessary uncertainty. As discussed in [6], “Point estimates alone can be misleading, as they do not quantify the variability or reliability of the estimate.” In particular , “confidence intervals (and by extension, bands) offer a way to convey the precision of estimates, reminding us that data are noisy and estimates are not exact”, and they “help us see how useful a model might be by explicitly recognising its limitations” as discussed in [7] and [8] respectively.
>
> [6] Casella, George, and Roger Berger. Statistical inference. CRC Press, 2024.
>
> [7] Freedman, David A. Statistical models: theory and practice. Cambridge University Press, 2009.
>
> [8] Box, George EP, and Norman R. Draper. Empirical model-building and response surfaces. John Wiley & Sons, 1987.
>
> We would be happy to add the details discussed in the above paragraphs (and more) to the paper if the reviewer feels that this would better help to explain and motivate the importance of our work.
>
>
> To the reviewer’s first point. In our view the distance transformation is not the only contribution of the paper, the others being the theory developed (including the new results i.e. Theorems 2.8 and A.4) and the emphasis on the choosing the optimal score transformation based on a learning dataset, (and justifying this theoretically) which as far as we are aware has not been suggested in the literature on conformal inference for images. This is important no matter what error rate is being controlled.
>
> However the distance transformation is an important part of the paper. It has the distinct advantage of allowing conformal sets to work very well where other existing approaches such as [2] fail, sometimes extremely badly (c.f. Figure A20). Fixing other existing methods in this manner is, in our view, an important contribution.
>
> [2] Mossina et al. Conformal Semantic Image Segmentation: Post-hoc Quantification of Predictive Uncertainty, CVPR Workshops, 2024.
>
> We look forward to hearing the reviewer's thoughts on our response and thank them once more for the time taken to read our work.

---

> > ### Comment · Reviewer_8vPV · 2024-11-29
> >
> > Thanks for the explanation. I understand the practical value of uncertainty quantification; however, I cannot see the practical value of uncertainty quantification in the way that the paper proposed. Perhaps, this is because the quantification is different than the uncertainy qualtification papers that I am more familiar with.
> >
> > The existing segmentation algorithms for uncertainty quantification, e.g. Probabilistic Unet, PhiSeg, and so on, return multiple possible segmentations. And, they demonstrate that the quality of uncertainty quantification by show that the distribution of the samples generated by the network and the experts is similar. They mostly use a metric called as generalized energy distance for quantification. In this setting, if the network is uncertain about an image, then the image can be delegated to human experts for detailed analysis or to get a consensus. This makes sense since they already show that the uncertainty measure of the network is similar to the experts.
> >
> > The paper does not present such an analysis and the authors responded my previous question about this as "In the case that experts disagree on the true segmented mask we would for now recommend using a consensus mask which is a function of the masks produced by each expert.". In this case, how can we make sure that the uncertainty quantified by the proposed method through the margin of the confidence bound reflects the true uncertainty. I would appreciate if the authors point me the specific table/figure where they present such quantification. This is extremely important because otherwise I cannot see a practical value of such uncertainty quantification. The use case previously mentioned by the authors, restricting the area of that the experts need to search for a certain structure, is not convincing to me because of the reasons I explained in my previous comment.
> >
> > One response of the authors that I might have overlooked before is the following "... as the predicted segmented mask approaches the ground truth mask in Hausdorff distance both inner and outer sets will converge to the ground truth mask." This could be actually an extremely important use case of the method if the negative correlation between the quantified uncertainty and the actual accuracy of the network (Hausdorff distance or Dice Score) is high. And, I believe this should be the main power that needs to be emphasized in the paper. The authors point a theorem and some visual results in their response but I couldn't really find analysis quantifying this. If there is a high correlation better than the other uncertainty quantification methods, such analysis makes the paper much stronger.
> >
> > Since I think these additional analysis require another round of major revision, I retain my current score.

---

> > > ### Author Response · Authors · 2024-12-03
> > >
> > > Thanks for your further thoughts. The uncertainty quantification we are describing is indeed quite different from the papers that the reviewer mentions. Those papers rely on probabilistic model assumptions which are not guaranteed to hold in practice. Instead conformal confidence sets provide robust guarantees which hold without making additional assumptions.
> > >
> > > With regards to the uncertainty quantification figures, we would like to point the reviewer to Figures 4, A21 and A25 in which we quantified the uncertainty and showed that the method provided the right guarantees. The width of the confidence sets are instead compared in Figures 5 and A17 and show big improvement over existing approaches whilst maintaining the same level of coverage.
> > >
> > > We are glad that the reviewer agrees that the new result which we have provided is an important contribution. This result shows that the distance transformed scores provide guarantees which cannot be provided by the untransformed scores. As shown in the brain imaging application the confidence sets provided by the untransformed scores can be very poor in practice while the distance transformed scores are very informative. There is indeed a negative correlation that the reviewer describes, compare e.g. the table on the segmentation performance in Section A.9 with the performance of the distance transformed scores. We would be happy to include a range of further simulations which illustrate this in further detail for the final version of the paper.

---

### Official Review · Reviewer_Bd8E · 2024-11-02

**Soundness:** 2
**Presentation:** 2
**Contribution:** 2
**Rating:** 5
**Confidence:** 3

**Summary:**

The authors formally present an approach that aims at inferring uncertainty margins to segmentations. They propose either take the logit score of a CNN and to threshold it to obtain this margin, or to threshold at a certain distance to the predicted segmentation. Threshold and type of margin (logit score / distance) is to be identified experimentally for a given dataset. Experiments on one public dataset  are shown (containing still images from minmally invasive surgery).

**Strengths:**

* The authors present the problem in a formal manner, relating it to existing work.
* The overall problem addressed is relevant.

**Weaknesses:**

* The motivation for the scores functions (logit, distance, ...) is weak. The necessity to choose the type and to even mix them gives the overall approach a bit of a heuristic touch. (While I do understand that you would consider your contribution here to be in the formal derivation of underlying theory, i.e., very much the opposite of a heuristic.)
* The experiments only provide insights into one very narrow application. they are merely fulfilling the purpose of an illustation of the problem, but not a validation.

**Questions:**

* You are testing on public data. Has your pretrained polyp segmentation algorithm been trained on the same public data?
* Are there any susequent video frames in the dataset, or images of the same polyp / patient? If there are, did you stratify your training / testing set accordingly?
* Please remove the reference to tumors throughout the paper. Polyps may be precursors to tumors, but they aren't any.
* You are using a dataset from different centers, there may be systematic differences in how the polyp areas are annotated - some annotators being more inclusive with respect to surrounding tissue, others being less. How does this variability impact on your measure?
*  I might have missed it but what is the accuracy of your underlying segmentation algorithm? I would be under the impression that it is a well performing algorithm on a rather easy segmentation task? How does your approach relate to extrema in algorithmic performance, i.e., perfect segmentations or complete misses?
* You are stating "In order to make efficient use of the data available, the learning dataset can in fact contain some or all of the data used to train the image segmentor." Your training data may be fairly overfittet impacting on your logit score and, hence, your choice of margin (logit/distance, thresholds). Wouldn't it be a safer approach to generate cross-validated logit functions and use them in the comparison?
* I understand that the primary contribution of this study is the theory offered. Still, you are stressing that your algorithm is a very lightweight addition to any pretrained segmentation algorithm. And there are a lot of standard computer vision / biomedical image data sets for segmentation available, as well as pretrained algorithms. Would you be able to generate segmentations maps for predefined certainty levels, and compare these levels with the testing performances across a larger set of applications? It would be quite convincing, if e.g., your 90% certainty map of the outer margin would indeed include 90% pixels of a test set or lead to a sufficiently large overlap (that has previously been defined) in 90% of all test cases.

---

> ### Author Response · Authors · 2024-11-20
>
> We are very grateful for the reviewer’s comments and remarks. Our response is below.
>
> Regarding the reviewers first concern. We in fact view the adaptability of the choice of score function to the dataset/model as a strength not a weakness of the method because different datasets/models have different features that mean the optimal score transformation may vary. This approach (of learning transformations on an independent dataset) has been previously used and theoretically justified in [1] in the context of conformal inference for time series data, in which the optimal copula was chosen based on a learning dataset (as we mention in Section 2.4). In the new datasets provided the optimal score transformations are different than for the polyps dataset and indeed certain choices (such as the original scores) can perform very badly (e.g. in the brain imaging application, see Figures A18 and A20 in the updated manuscript). This helps to illustrate the need to optimize the score functions.
>
> [1] Sun, Sophia, and Rose Yu. "Copula conformal prediction for multi-step time series forecasting." ICLR, 2024.
>
> We indeed regard one of the main contributions to be the theory developed. We would like to clarify that learning the score functions on an independent learning is theoretically valid. Crucially, as for the similar approach taken in [1] in the time series setting, the independence of the learning data set from the calibration and testing datasets guarantees the vaildity of the optimally chosen score function.
>
> Regarding the reviewers second concern that we only consider a single dataset we thank the reviewer for this comment. In order to address this we have added two additional datasets involving brain imaging and teeth segmentation. These new datasets help to illustrate the robustness and usefulness of our method.
>
> Regarding the reviewers questions.
>
> 1- The dataset is indeed public but is independent of the data used to train the original polyp segmentation model.
>
> 2- In the original dataset there were a few images from the same video frames however we removed these for the purposes of our analysis. This is important our model assumes exchangeability which would be violated if there was dependence between the images. We now clarify in Section 3 that the images used come from different patients.
>
> 3- Regarding the use of the word tumor, we apologize for this oversight and have mostly removed the word throughout the paper, except in one setting in which we are not referring to polyps in particular. We thank the reviewer for pointing this out.
>
> 4- The reviewer is right to note that the fact that the data is from different centres may influence the annotations. We rely strongly on a good quality ground truth and the model is only as good as the ground truth available. Where possible we would recommend taking a consensus rating by combining the annotations of multiple annotators and then using this consensus in combination with our model for best results.
>
> 5 – We have now included a table in Appendix A.9 illustrating the performance of the different segmentation models used, measured in terms of dice, precision and recall scores. This table helps to show how improvements in these metrics correspond to improvements in the performance of our method. In particular for the best performing model based on these metrics (the HDBET model designed for brain extraction), has relatively tight confidence sets. This is a relationship which we have now formalized in Theorems 2.8 and A.4 results which give guarantees on the size of the resulting confidence sets related to the performance of the model. Other choices of score function do not correlate with these metrics, indeed the original untransformed scores perform notably badly for the brain imaging application despite the high performance on the metrics (see e.g. Figure A20). As such, even for well performing segmentation models, appropriate score transformations are required in order to obtain tight confidence bounds. Theorems 2.8 and A.4 show that the model handles perfect segmentations very well as they have a Hausdorff distance of 0 from the true mask. Instead complete misses will typically increase the size of the confidence bands, appropriately so as they indicate a failure of the model in that instance.

---

> ### Author Response · Authors · 2024-11-20
>
> 6- We agree that using the learning dataset the training data may not provide the optimal score transformations. This is a not a problem for validity as the training data is assumed to be independent of the calibration and test datasets and so the results of Sections 2.2 and 2.3 still ensure that we can make inclusion statements with confidence. However doing so may impact the choice of score functions. As we now take greater care to emphasise in Section 2.4, we do not recommend using the training data as part of the learning dataset if there is a large amount of data available. However in cases where there is limited data (such as the teeth segmentation problem which we now consider), learning the score function on the training data may still be helpful. In particular the training data may still contain information which allow us to distinguish between different score functions. We saw in the application to teeth segmentation in which the score functions on the learning dataset (which was made up of the training data) had a similar performance on the test dataset. Doing so means that we are not required to give up any of the calibration data used, or make the decision to train the model using fewer images. This is a trade off that must be decided upon carefully by the researcher and where possible we recommend that researchers use a learning dataset which is independent of the training data (as we do with the polyps and brain imaging data settings which we consider).  Where possible we thus recommend splitting the data into independent training, learning, calibration and test datasets.
>
> 7- Our method is indeed a lightweight addition to any existing black box image segmentation model and is relatively easy to apply to additional datasets. The new datasets and applications which we have added to the paper help to illustrate this, showing that the model is generally applicable, informative and valid in these settings. In particular, for each of the models considered, we perform validations in which we resample with replacement from the data in order to check the coverage rate of the method, see Sections 3.3, A.7.4 and A.8.4 of the updated manuscript. We would like to clarify that the guarantees are in fact that 100% of true mask is included 90% of the time rather than that 90% of the mask is included 90% of the time. This guarantee allows full coverage and means that the resulting confidence sets are more meaningful.  We shall include validations across additional settings for the camera ready version of the paper.
>
> We thank the reviewer once more for their helpful comments and look forward to hearing their thoughts on our response and discussing any follow up questions that they may have.

---

> ### Author Response · Authors · 2024-11-25
>
> Dear Reviewer Bd8E,
>
> We were wondering whether you had had a chance to have a look at the new changes which we have made to the paper in our response. In our view the new version of the paper has been greatly improved thanks to your review. Do you feel that we have addressed your comments?
>
> Thanks in advance!

---

> > ### Comment · Reviewer_Bd8E · 2024-11-26
> > **Still rather unhappy with the global score functions**
> >
> > Thank you for your reply. And let me reiterate that I can see that your contribution is a methodological one. However, the concept of relying on rather simple and global transformations - distance or threshold adjustment - seems to me to be too simplistic in terms of practical relevance. Let me rephrase why I find your approach of choosing score functions limiting:
> >
> > In most biomedical segmentation tasks, you would have sharp decisions in some parts of the foreground boundary - just because there is a clearly visible difference between foreground and background - and in other areas you would have a rather smooth transition between "very likely foreground" and "very unlikely foreground".  An example is your brain extraction map: the difference between brain and skull at the top of the head is clearly visible. The segmentation performance of most algorithms is likely to be accurate at the pixel level. Other parts of the outer boundary of the brain are not so clearly separated in terms of image intensities (such as the base of the brain), or are simply invisible and the result of anatomical reasoning extrapolating a general shape (such as the boundary between the brain and the brainstem).  If my algorithm is producing segmentation errors for whatever reason, and I am still unhappy with the segmentation performance after adjusting the threshold on the logit, my only option is to add a few millimetres of uncertainty to the boundary everywhere. In many applications where the foreground is surrounded by a rather inhomogeneous background, or where the shape of the foreground is defined in a somewhat inconsistent way (by experts, image intensity differences, etc.), this is unlikely to be a valid assumption for how errors and uncertainties are distributed around the boundary of your foreground.  [And all this reasoning also applies to the uncertainty of the inward direction of an inhomogeneous foreground].
> >
> > So, while I appreciate your methodological contribution, I feel that some of the basic assumptions underlying your solution are not very convincing to me. You would convince me with an empirical study, on a variety of data sets, demonstrating that inter-related "logit-distance" scores are ok to be ignored, in favor of using pure "logit" or "distance" scores only. But I see that a rebuttal is not the right place for this extra effort.

---

> > > ### Author Response · Authors · 2024-12-03
> > >
> > > We would like to thank the reviewer for their further thoughts. We agree that one of the main contributions is methodological however they are practical as well. I.e. the distance transformations which we propose are also the difference between the methods working and not working (see e.g. Figure A20).
> > >
> > > We're grateful for the thoughts of the reviewer regarding the brain imaging application. We would like to clarify that given the time available for the additional analyses we were not able to explore all possible choices of score transformations. Moreover it would in fact be possible to explore score transformations which take the shape/point of the brain into account and adjust accordingly - allowing for a more variable width across the brain. We would be happy to explore this for the final version of the paper.  While it is indeed natural to add millimetres of uncertainty, doing so conformally is key in order to provide valid inference.

---

### Official Review · Reviewer_Jpsi · 2024-11-04

**Soundness:** 3
**Presentation:** 3
**Contribution:** 2
**Rating:** 6
**Confidence:** 2

**Summary:**

Authors develop confidence sets providing spatial uncertainty guarantees for outputs of a black-box machine learning model designed for image segmentation. Specifically, this paper adapts conformal inference to the imaging setting, obtaining thresholds on a calibration dataset based on the distribution of the maximum of the transformed logit scores within and outside of the ground truth masks. Qualitative evaluations are implemented on a polyp tumor dataset to demonstrate the effectiveness of this approach.

**Strengths:**

1.  The topic of this work is quite interesting. By proposing the concept of conformal confidence sets, this work could provide spatial uncertainty guarantees for the outputs of image segmentation models.
2.  Theoretical proofs are well formulated to serve as a strong proof for this paper.

**Weaknesses:**

1.  A very obvious typos “polpys” exist many times, even in the abstract. That should be “polyps”.
2.  It will be more convincing if authors could provide quantitative results for the segmentation performance of polyp segmentation. The evaluation metrics include Dice, Precision, Recall, etc. For comparable baseline models, authors could choose PraNet, SANet, etc.
3.  Since the concept of conformal confidence sets can be generalized to other medical image segmentation tasks, maybe more public datasets are applicable to this work, such as vertebrae or tooth segmentation.
4.  Some technical terms need to be further explained for a better understanding, such as FWER/FDR/FDP in the introduction part.

**Questions:**

Please refer to the weakness part.

---

> ### Author Response · Authors · 2024-11-20
>
> We are very grateful for the reviewer’s comments and remarks. We agree that providing uncertainty quantification for black box neural network models is an interesting problem.
>
> Regarding additional datasets we have taken the reviewer’s advice on board and have now included extensive analysis of two further datasets. The first is a brain imaging dataset. And, the second, following the reviewer’s suggestion, involves teeth segmentation. As now shown in the main text (Sections 4 and 5) our method works very well in these scenarios providing meaningful confidence sets which have robust confidence guarantees. This demonstrates that our method extends robustly to other settings and models. Further analysis is shown in Sections A.7 and A.8.
>
> We also now included Dice, Precision and Recall metrics (evaluated on the corresponding validation dataset) for each of the 3 segmentation algorithms considered. (I.e. PraNet, HDBET and the UNET based GAN model we used for teeth segmentation). See the relevant table Section A.9 in the updated draft for full details. The results are very helpful in understanding how to performance of the models affects the performance of the confidence sets. In particular improvements in these metrics correspond to improvements in the quality of the confidence sets based on the distance transformed scores. The HDBET model has the highest dice score and has the tightest confidence sets as a result. Note that other score transformations such as the identity (which yields the original logit scores) do not have this monotonicity property. Indeed Figure A20 shows that the untransformed logit scores can be very uninformative, but the degree to which this is true depends on the application. In order to formalize the relationship between the distance transformed scores and the quality of the model we have provided new results (Theorems 2.8 and A.4) which further motivate the use of the distance transformation. Comparison of between the metrics now shown in Section A.9 and the performance of the confidence sets helps to illustrate this result.
>
> Regarding the comparison to other baseline models, we shall include measures of the performance (e.g. relative to SANet and UACAnet and others) in the final version of the manuscript.
>
> Regarding the technical terms (FWER/FDR) we have fully written out their acronyms for clarity where they are introduced and have included a new section of the Appendix (Section A.10) in which these are formally defined and where we discuss the relationship between them and different measures of coverage in the segmentation setting.
>
> We would like to apologize for the spelling error of polyps which we have now corrected in the updated draft and we thank the reviewer for pointing this out.
>
> We thank the reviewer once more for their helpful comments and look forward to hearing their thoughts on our response and discussing any follow up questions that they may have.

---

> ### Author Response · Authors · 2024-11-25
>
> Dear Reviewer Jpsi,
>
> We were wondering whether you had had a chance to have a look at the new changes which we have made to the paper in our response. In our view the new version of the paper has been greatly improved thanks to your review. Do you feel that we have addressed your comments?
>
> Thanks in advance!

---

### Author Response · Authors · 2024-11-20

We would like to thank the reviewers for their feedback and constructive comments and for taking the time to read our work. All reviewers stressed the need to apply our methods to more than one dataset. We have taken this feedback to heart and in order to address this we now include the results of applying the methods to 2 new datasets and problems: brain mask segmentation and teeth segmentation. We find in these settings that the method works well, providing informative inner and outer confidence sets. On these datasets we also explored the impact of alternative score transformations based on smoothing the original untransformed scores with a kernel of varying bandwidth. The results of these applications have been included as Sections 4 and 5 of the manuscript, with comparisons between different transformations including smoothing included in Sections A.7 and A.8.

On these datasets, as previously with the polyps data, the best combination of score transformations was learnt from a independent learning dataset. For brain mask segmentation the distance transformed scores provided the tightest regions for both inner and outer confidence sets whilst the original (untransformed scores) were uninformative. Smoothing improved the original scores but not as much as applying the distance transformation. Instead for teeth segmentation distance transformed scores provided informative outer sets whilst smoothing the untransformed scores provided the best inner sets. Since the best transformation depends on the application these new data applications help to illustrate the importance of learning the score function in this manner.

We have also included new results (Theorems 2.8 and A.4) which characterise the relationship between the confidence sets based on the distance transformed scores and the Hausdorff distance between predicted and ground truth masks on the calibration dataset. These results shows that if the Hausdorff distance between predicted and ground truth masks on the calibration sets is bounded then confidence sets for new observations are guaranteed to be at most twice as wide as the bound. Importantly a corresponding result does not hold for the untransformed scores as we illustrate Figure A20. A comparison of the metrics, of the segmentation models used, is now included in Section A.9 and is related to the performance of the distance transformed scores (i.e. the model (HDBET) with the highest performance on these metrics has the most precise confidence bands). We have also added a description of the algorithm at test time in Section A.5.

We have uploaded a new version of the paper with the results of applying our method to these datasets and other changes in response to the reviewers comments. Changes in this new version are shown in red and sections referred to in the responses below refer to sections of the newly uploaded paper.

---

### Meta-Review · Area_Chair_FeTS · 2024-12-20

**Metareview:**

1. The method is only evaluated on a single dataset and focuses exclusively on binary segmentation.
2. The manuscript lacks a thorough comparison with existing conformal prediction methods for image segmentation.
3. Some aspects of the methodology, particularly how the algorithm operates during testing and how uncertainty is quantified, require clearer explanation to ensure readers can fully understand and reproduce the results.
4. Reviewers highlighted the need for more quantitative results, including performance metrics like Dice, Precision, and Recall for segmentation, as well as aggregated coverage scores.

**Additional Comments On Reviewer Discussion:**

The paper received mixed reviews and the authors were able to address some of the concerns raised by the reviewers. While there is no final consensus, the AC acknowledges both the merits outlined by the positive reviewers & shortcomings of the paper. The discussion clarified many of the concerns, however, after reading all discussions and responses, it appears that the paper requires a major revision to meet the standards required by ICLR and requested by the reviewers, e.g., the practical application, and more quantitative results.
This means that the paper cannot be accepted in its current form and needs to be reviewed again potentially in a future venue!

---

### Decision · Program_Chairs · 2025-01-22

Reject